# Exploratory Diffusion Model for Unsupervised Reinforcement Learning

**Chengyang Ying**[1]**, Huayu Chen**[1]**, Xinning Zhou**[1]**, Zhongkai Hao**[1]**,**
**Hang Su**[1,*]**, Jun Zhu**[1,*]
[1]Department of Computer Science & Technology, Institute for AI, BNRist Center,
Tsinghua-Bosch Joint ML Center, THBI Lab, Tsinghua University

## Abstract

Unsupervised reinforcement learning (URL) pre-trains agents by exploring diverse states in reward-free environments, aiming to enable efficient adaptation to various downstream tasks. Without extrinsic rewards, prior methods rely on intrinsic objectives, but heterogeneous exploration data demand strong modeling capacity for both intrinsic reward design and policy learning. We introduce the **Ex**ploratory **D**iffusion **M**odel (**ExDM**), which leverages the expressive power of diffusion models to fit diverse replay-buffer distributions, thus providing accurate density estimates and a score-based intrinsic reward that drives exploration into under-visited regions. This mechanism substantially broadens state coverage and yields robust pre-trained policies. Beyond exploration, ExDM offers theoretical guarantees and practical algorithms for fine-tuning diffusion policies under limited interactions, overcoming instability and computational overhead from multi-step sampling. Extensive experiments on Maze2d and URLB show that ExDM achieves superior exploration and faster downstream adaptation, establishing new state-of-the-art results, particularly in environments with complex structure or cross-embodiment settings. The source code is provided at https://github.com/yingchengyang/ExDM.

## 1 Introduction

Developing agents that generalize across diverse tasks remains a central challenge in reinforcement learning (RL). Unsupervised RL (URL) (Eysenbach et al., 2018; Laskin et al., 2021) aims to address this by pre-training in reward-free environments to acquire diverse skills or transferable representations. In the absence of extrinsic rewards, agents often rely on intrinsic objectives that are frequently hand-crafted, myopic, and weakly aligned with downstream tasks. The data collected through exploration are highly heterogeneous, demanding representations that are expressive yet stable against collapse or spurious correlations. In addition, policies trained in fixed reward-free settings often fail to transfer under shifts in dynamics, embodiment, or semantics.

A central obstacle in URL is the demand for strong modeling capacity during both pre-training and fine-tuning. Effective exploration in reward-free environments hinges on intrinsic rewards derived from accurate estimates of the underlying state distribution, which is typically heterogeneous and difficult to capture. Existing methods can collect diverse trajectories but often rely on simple pre-trained policies—such as Gaussian (Pathak et al., 2017; Mazzaglia et al., 2022) or discrete skill-based policies (Eysenbach et al., 2018; Laskin et al., 2022)—chosen for their ease of training and sampling. Such policies fail to capture the full diversity of explored data in the replay buffer, limiting both unsupervised exploration and downstream adaptation. This calls for more powerful modeling approaches, where diffusion models stand out for their stability and strong density estimation ability.

To address these challenges, we propose the **Ex**ploratory **D**iffusion **M**odel (**ExDM**), which leverages diffusion-based density estimation to address the exploration bottleneck in unsupervised RL while providing a reusable prior for downstream adaptation. At its core, ExDM trains a diffusion model on the heterogeneous and nonstationary state distribution in the replay buffer, and uses the resulting

---

*H. Su and J. Zhu are corresponding authors.

Figure 1: **Overview of Exploratory Diffusion Model (ExDM).** Different from standard RL, URL aims to explore in reward-free environments, requiring expressive policies and models to fit heterogeneous data (Theorem 4.1). During pre-training, ExDM employs the diffusion model to model the heterogeneous exploration data and calculate score-based intrinsic rewards to encourage exploration. Moreover, we adopt a Gaussian behavior policy to collect data that avoids the inefficiency caused by the multi-step sampling of the diffusion policy.

score function to define an intrinsic reward $\mathcal{R}_{\text{score}}$ that explicitly targets under-visited states. This drives broad state coverage and maximizes entropy during reward-free exploration. Unlike conventional generative uses of diffusion, ExDM employs the score function for exploration utility rather than sample fidelity, and must learn from an online replay buffer with shifting visitation distributions. To keep train efficiency, ExDM decouples modeling from acting—replacing costly multi-step diffusion sampling with a lightweight Gaussian policy trained to maximize $\mathcal{R}_{\text{score}}$. This design preserves the modeling strength of diffusion while enabling scalable training and action selection.

Beyond enhancing unsupervised exploration, ExDM also provides a strong initialization for downstream tasks. In addition to fine-tuning the Gaussian behavior policy with standard RL algorithms, ExDM allows the diffusion model itself to be adapted for downstream control. This adaptation is particularly challenging in URL, where fine-tuning must succeed with limited online interaction. To this end, we analyze the fine-tuning objective and derive an alternating optimization procedure whose convergence and optimality are formally established in Theorem 4.2.

We evaluate ExDM on both unsupervised exploration and downstream adaptation across standard benchmarks, including Maze2d (Campos et al., 2020) and continuous control in URLB (Laskin et al., 2022). In Maze2d, ExDM consistently achieves substantially higher state coverage than all baselines. On the most challenging mazes with many branching paths and decision points, ExDM attains up to **51% higher coverage** and reaches comparable performance using only **37% of timesteps**, demonstrating its ability to efficiently explore diverse regions under strict interaction budgets, while baselines often stall near corners and fail to cover the maze. Beyond exploration, URLB experiments, including single-embodiment and cross-embodiment settings, further show that ExDM adapts rapidly to diverse downstream tasks and **outperforms SOTA URL and diffusion fine-tuning baselines by large margins**, highlighting its effectiveness as a general framework for exploration and transfer.

In summary, the main contributions are as follows:

- To the best of our knowledge, this is the first work to introduce diffusion models into unsupervised RL, enabling accurate modeling of heterogeneous state distributions and defining a score-based intrinsic reward that substantially improves exploration.

- Beyond exploration, ExDM develops an efficient decoupled training scheme and a fine-tuning algorithm for adapting pre-trained diffusion components to downstream tasks under limited interaction, with theoretical guarantees of convergence and optimality.

- Extensive experiments on Maze2d and URLB benchmarks demonstrate that ExDM achieves broader state coverage and faster adaptation than prior methods, establishing new state-of-the-art performance in both exploration and transfer.

## 2 RELATED WORK

**Unsupervised Pre-training in RL.** For achieving zero-shot generalization in RL, there are lots of attempts like in-context RL (Rakelly et al., 2019; Zintgraf et al., 2021) or forward-backward repre-

sentations (Touati & Ollivier, 2021; Tirinzoni et al., 2025). However, these methods require sampling from tasks during pre-training or reward signals from the offline datasets. Differently, URL pre-trains agents in reward-free environments to acquire knowledge for fast fine-tuning downstream tasks. Existing methods mainly rely on intrinsic rewards to guide agents to explore the environment, falling into two categories: exploration and skill discovery. Exploration methods typically explore diverse states by maximizing intrinsic rewards designed to estimate either uncertainty (Pathak et al., 2017; Burda et al., 2018; Pathak et al., 2019; Raileanu & Rocktäschel, 2020; Mazzaglia et al., 2022; Yuan et al., 2023; Ying et al., 2024) or state entropy (Lee et al., 2019; Liu & Abbeel, 2021; Seo et al., 2021; Mutti et al., 2021). Skill-discovery methods hope to collect diverse skills by maximizing the mutual information between skills and states (Eysenbach et al., 2018; Lee et al., 2019; Campos et al., 2020; Kim et al., 2021; Park et al., 2022; Laskin et al., 2022; Yuan et al., 2022; Zhao et al., 2022; Yang et al., 2023b; Park et al., 2023; Bai et al., 2024; Wilcoxson et al., 2024). Although exploring diverse states, existing methods always neglect the expression ability of pre-trained policies and choose simple Gaussian policies (Pathak et al., 2017; Mazzaglia et al., 2022) or skill-based policies (Eysenbach et al., 2018; Yang et al., 2023b), which fail to capture the diversity present in the explored data. Consequently, applying generative models with strong expressive ability for improving the diversity of pre-trained policies is still less studied.

**RL with Diffusion Models.** Recent advancements have shown that high-fidelity diffusion models can benefit RL from different perspectives (Zhu et al., 2023). In offline RL, diffusion policies (Wang et al., 2023; Chen et al., 2023; Lu et al., 2023; Chi et al., 2023; Hansen-Estruch et al., 2023; Kang et al., 2024) excel at modeling multimodal behaviors, outperforming previous policies such as Gaussians. Besides policies, diffusion planners (Janner et al., 2022; Ajay et al., 2023; He et al., 2023; Liang et al., 2023; Nuti et al., 2023; Chen et al., 2024a) have demonstrated the potential in long-term sequence prediction and test-time planning. Some works have also investigated online training diffusion policies to improve performance (Psenka et al., 2023; Li et al., 2024; Ren et al., 2024; Mark et al., 2024; Celik et al., 2025; Ma et al., 2025; Ishfaq et al., 2025). However, the computational cost of multi-step sampling remains the efficiency bottleneck. In addition to behavior modeling, diffusion models have also been employed as world models (Alonso et al., 2024; Ding et al., 2024), augmented replay buffer (Lu et al., 2024; Wang et al., 2024a), hierarchical RL (Li et al., 2023; Chen et al., 2024b), and so on. To the best of our knowledge, this work represents the first attempt to leverage the strong modeling capabilities for heterogeneous distribution of diffusion models for unsupervised exploration.

## 3 BACKGROUND

### 3.1 UNSUPERVISED REINFORCEMENT LEARNING

RL considers Markov decision processes (MDP) $\mathcal{M} = (\mathcal{S}, \mathcal{A}, \mathcal{P}, \mathcal{R}, \rho_0, \gamma)$. Here $\mathcal{S}$ and $\mathcal{A}$ denote the state and action spaces, respectively. For $\forall (\boldsymbol{s}, \boldsymbol{a}) \in \mathcal{S} \times \mathcal{A}$, $\mathcal{P}(\cdot | \boldsymbol{s}, \boldsymbol{a})$ is a distribution on $\mathcal{S}$, representing the dynamic of $\mathcal{M}$, and $\mathcal{R}(\boldsymbol{s}, \boldsymbol{a})$ is the extrinsic task reward function. $\rho_0$ is the initial state distribution and $\gamma$ is the discount factor. For a given policy $\pi : \mathcal{S} \rightarrow \Delta(\mathcal{A})$, we define the discount state distribution of $\pi$ at state $\boldsymbol{s}$ as $d_\pi(\boldsymbol{s}) = (1 - \gamma) \sum_{t=0}^{\infty} [\gamma^t \mathcal{P}(\boldsymbol{s}^t = \boldsymbol{s})]$. The objective of RL is to maximize the expected cumulative return of $\pi$ over the task $\mathcal{R}$:

$$J(\pi) \triangleq \mathbb{E}_{\tau \sim \mathcal{M}, \pi} \left[ \mathcal{R}(\tau) \right] = \frac{1}{1 - \gamma} \mathbb{E}_{\boldsymbol{s} \sim d_\pi, \boldsymbol{a} \sim \pi} \left[ \mathcal{R}(\boldsymbol{s}, \boldsymbol{a}) \right]. \tag{1}$$

To boost agents' generalization, unsupervised RL (URL) typically includes two stages: *unsupervised pre-training* and *few-shot fine-tuning*. During pre-training, agents explore the reward-free environment $\mathcal{M}^c$, i.e., $\mathcal{M}$ without the reward function $\mathcal{R}$. Thus, URL requires designing intrinsic rewards $\mathcal{R}_{\text{int}}$ to guide policies to maximize the state entropy $\mathcal{H}(d_\pi(\cdot))$. During fine-tuning, agents adapt pre-trained policies to handle downstream tasks represented by extrinsic task-specific rewards $\mathcal{R}$, through limited interactions (like one-tenth of pre-training steps, the formulation is in Eq. 9).

### 3.2 DIFFUSION MODELS IN REINFORCEMENT LEARNING

Recent studies have demonstrated that diffusion models (Sohl-Dickstein et al., 2015; Ho et al., 2020) excel at accurately representing heterogeneous behaviors in continuous control, particularly through

the use of diffusion policies (Wang et al., 2023; Chi et al., 2023). Given state-action pairs $(\boldsymbol{s}, \boldsymbol{a})$ sampled from some unknown policy $\mu(\boldsymbol{a}|\boldsymbol{s})$, diffusion policies consider the forward diffusion process that gradually injects standard Gaussian noise $\boldsymbol{\epsilon}$ into actions:

$$\boldsymbol{a}_t = \alpha_t \boldsymbol{a} + \sigma_t \boldsymbol{\epsilon}, \quad t \in [0, 1], \tag{2}$$

here $\alpha_t, \sigma_t$ are pre-defined hyperparameters satisfying that when $t = 0$, we have $\boldsymbol{a}_t = \boldsymbol{a}$, and when $t = 1$, we have $\boldsymbol{a}_t \approx \boldsymbol{\epsilon}$. For $\forall t \in [0, 1]$, we can define the marginal distribution of $\boldsymbol{a}_t$ as

$$p_t(\boldsymbol{a}_t|\boldsymbol{s}, t) = \int \mathcal{N}(\boldsymbol{a}_t|\alpha_t \boldsymbol{a}, \sigma_t^2 \boldsymbol{I}) \mu(\boldsymbol{a}|\boldsymbol{s}) d\boldsymbol{a}. \tag{3}$$

Then we train a conditional "noise predictor" $\boldsymbol{\epsilon}_\theta(\boldsymbol{a}_t|\boldsymbol{s}, t)$ to predict the added noise of each timestep:

$$\min_\theta \mathbb{E}_{t, \boldsymbol{\epsilon}, \boldsymbol{s}, \boldsymbol{a}}[\|\boldsymbol{\epsilon}_\theta(\boldsymbol{a}_t|\boldsymbol{s}, t) - \boldsymbol{\epsilon}\|^2]. \tag{4}$$

The learned $\boldsymbol{\epsilon}_\theta$ can estimate the score function $\nabla_{\boldsymbol{a}_t} \log p_t(\boldsymbol{a}_t|\boldsymbol{s}_t, t)$. We can discretize diffusion ODEs of the reverse process (Song et al., 2021b) and sample actions with numerical solvers (Song et al., 2021a; Lu et al., 2022) in around $5 \sim 15$ steps, to approximate the original policy $\mu(\boldsymbol{a}|\boldsymbol{s})$. However, this multi-step sampling affects the training efficiency, especially in online settings.

## 4 METHODOLOGY

Below, we introduce the Exploratory Diffusion Model (ExDM) to capture diverse data to boost unsupervised exploration (Sec. 4.1) and obtain powerful initialization for fast fine-tuning (Sec. 4.2).

### 4.1 EXPLORATORY DIFFUSION MODEL FOR UNSUPERVISED PRE-TRAINING

The major challenge and objective during unsupervised pre-training is to explore diverse states in reward-free environments. Consequently, a natural pathway is to pre-train the policy to maximize the entropy of the state (Liu & Abbeel, 2021), i.e., $\mathcal{H}(d_\pi(\cdot)) = \int_s -d_\pi(s) \log d_\pi(s) ds$. Although the optimal policy of fully-observable single-agent RL is a simple deterministic policy, we prove that, even if the environment is discrete, policies with the maximum state entropy are still complicated and not deterministic with a high probability, requiring much stronger modeling abilities.

**Theorem 4.1** (Policy with maximal state entropy). *When $\mathcal{S}, \mathcal{A}$ are discrete spaces, i.e., $|\mathcal{S}| = S, |\mathcal{A}| = A$, there are $M \triangleq A^S$ deterministic policies. Set $\hat{\pi} = \arg\max_\pi \mathcal{H}(d_\pi(\cdot))$, under some mild assumptions, we have*

$$P(\hat{\pi} \text{ is not deterministic policy and } \mathcal{H}(d_{\hat{\pi}}) = \log|S|) \geq 1 - M^S v(S)^M, \tag{5}$$

*will fast converge to 1 with the increasing of A, and here $v(S)$ is a constant only related to S and satisfies $0 < v(S) < 1$.*

Details and proof are in Appendix B.1 (we also discuss continuous situations there). This theorem demonstrates that maximizing state entropy requires policies with strong expression abilities, rather than simple deterministic policies. Despite previous work mainly considering simple Gaussian policies or skill-based policies, in practice, explored replay buffer is always diverse and heterogeneous, as the policy continuously changes to visit new states during pre-training. Consequently, URL requires capturing the heterogeneous distribution of collected data and obtaining policies with high diversity. These challenges pose the requirement of strong density estimation and fitting abilities, while maintaining training stability and efficiency. Inspired by the recent great success of diffusion models in modeling diverse image distributions (Dhariwal & Nichol, 2021) and behaviors (Chi et al., 2023; Janner et al., 2022), ExDM proposes to utilize the diffusion models $\boldsymbol{\epsilon}_{\theta'}$ and $\boldsymbol{\epsilon}_\theta$ to model the distribution of states and state-action pairs in the replay buffer $\mathcal{D}$ collected before:

$$\min \mathbb{E}_{\boldsymbol{s}, \boldsymbol{a} \sim \mathcal{D}}[\mathbb{E}_{t, \boldsymbol{\epsilon}}\|\boldsymbol{\epsilon}_{\theta'}(\boldsymbol{s}_t|t) - \boldsymbol{\epsilon}\|^2 + \mathbb{E}_{t, \boldsymbol{\epsilon}}\|\boldsymbol{\epsilon}_\theta(\boldsymbol{a}_t|\boldsymbol{s}, t) - \boldsymbol{\epsilon}\|^2]. \tag{6}$$

To maximize the entropy of the state distribution, we can use $\log p_{\theta'}(\boldsymbol{s})$ to measure the frequency of states in the replay buffer. Consequently, we design $-\log p_{\theta'}(\boldsymbol{s})$ as the intrinsic reward to encourage the agent to explore these regions. Although estimating the log-probability of the diffusion model

---

**Algorithm 1** Pre-training of ExDM

---

**Require:** Reward-free environment $\mathcal{M}^c$, replay buffer $\mathcal{D}$, Gaussian behavior policy $\pi_{\mathrm{g}}$, diffusion policy $\pi_{\mathrm{d}}$ parameterized with the score model $\boldsymbol{\epsilon}_\theta$, state diffusion model $\boldsymbol{\epsilon}_{\theta'}$.

1: **for** sample step $= 1, 2, ..., S$ **do**
2:     **for** update step $= 1, 2, ..., U$ **do**
3:         Sample $\boldsymbol{s}$-$\boldsymbol{a}$ pairs $\{(\boldsymbol{s}^m, \boldsymbol{a}^m)\}_{m=1}^M$ from $\mathcal{D}$.
4:         Update $\boldsymbol{\epsilon}_\theta$ and $\boldsymbol{\epsilon}_{\theta'}$ via optimizing with Eq. (6) with sampled data.
5:         Calculate score-based intrinsic rewards $\boldsymbol{r}^m$ via Eq. (8) for each sampled pair $(\boldsymbol{s}^m, \boldsymbol{a}^m)$.
6:         Train $\pi_{\mathrm{g}}$ with $(\boldsymbol{s}^m, \boldsymbol{a}^m, \boldsymbol{r}^m)$ by any off-policy RL algorithm.
7:     **end for**
8:     Utilize the behavior policy $\pi_{\mathrm{g}}$ to interact with $\mathcal{M}^c$ and store state-action pairs into $\mathcal{D}$.
9: **end for**

---

is challenging, it is well known that $-\log p_{\theta'}(\boldsymbol{s})$ can be bounded by the following evidence lower bound (ELBO) (Ho et al., 2020):

$$-\log p_{\theta'}(\boldsymbol{s}) \le \mathbb{E}_{\boldsymbol{\epsilon},t}[\boldsymbol{w}_t \|\boldsymbol{\epsilon}_{\theta'}(\boldsymbol{s}_t|t) - \boldsymbol{\epsilon}\|^2] + C, \tag{7}$$

here $C$ is a constant independent of $\theta'$, and $\boldsymbol{w}_t$ are parameters related to $\alpha_t, \sigma_t$, which are typically ignored (Ho et al., 2020). Consequently, we propose our score-based intrinsic rewards as:

$$\mathcal{R}_{\mathrm{score}}(\boldsymbol{s}) = \mathbb{E}_{\boldsymbol{\epsilon},t}[\|\boldsymbol{\epsilon}_{\theta'}(\boldsymbol{s}|t) - \boldsymbol{\epsilon}\|^2]. \tag{8}$$

Intuitively, our score-based intrinsic rewards can measure the fitting quality of the diffusion model to the explored data, thereby encouraging the agent to explore regions that are poorly fitted or unexplored (more analyses between $\mathcal{R}_{\mathrm{score}}$ and $-\log p_{\theta'}$ are in Appendix C.1). By maximizing these intrinsic rewards, ExDM trains agents to discover unseen regions effectively. However, directly using diffusion policies to interact with reward-free environments during pre-training is inefficient and unstable due to the requirement of multi-step sampling. To address this limitation, ExDM incorporates a Gaussian behavior policy $\pi_{\mathrm{g}}$ for efficient action sampling. Gaussian behavior policy $\pi_{\mathrm{g}}$ can be trained using any off-policy RL algorithm, guided by score-based intrinsic rewards $\mathcal{R}_{\mathrm{score}}(\boldsymbol{s})$. This encourages the exploration of regions where the diffusion model either fits poorly or has not yet been exposed. The pseudo code of the unsupervised exploration stage of ExDM is in Algorithm 1.

## 4.2 Efficient Online Fine-tuning to Downstream Tasks

When adapting pre-trained policies to downstream tasks with limited timesteps, existing URL methods always directly apply online RL algorithms like DDPG (Lillicrap, 2015) or PPO (Schulman et al., 2017) for fine-tuning. The behavior policy $\pi_{\mathrm{g}}$ in ExDM can also be fine-tuned to handle the downstream task with the same online RL algorithms, performing fair comparison of exploration efficiency between ExDM and baselines (detailed experimental results are in Sec. 5.3).

Besides $\pi_{\mathrm{g}}$, ExDM has also pre-trained the diffusion policy $\pi_{\mathrm{d}}$, which can better capture the heterogeneous explored trajectories for adapting to downstream tasks. Unfortunately, it is challenging to online fine-tune diffusion policies due to the instability caused by the multi-step sampling and the lack of closed-form probability calculation (Ren et al., 2024). To address these challenges, we first analyze the online fine-tuning objective for URL. Given the limited fine-tuning timesteps, the objective can be formulated as the combination of maximizing the cumulative return and keeping close to the pre-trained policy over all $\boldsymbol{s}$ (Eysenbach et al., 2021) (more analyses are in Appendix C.2):

$$
\begin{aligned}
\max_\pi J_{\mathrm{f}}(\pi) &\triangleq J(\pi) - \frac{\beta}{(1-\gamma)}\mathbb{E}_{\boldsymbol{s}\sim d_\pi}\left[D_{\mathrm{KL}}(\pi(\cdot|\boldsymbol{s})\|\pi_{\mathrm{d}}(\cdot|\boldsymbol{s}))\right] \\
&= \frac{1}{1-\gamma}\mathbb{E}_{\boldsymbol{s}\sim d_\pi, \boldsymbol{a}\sim\pi}\left[\mathcal{R}(\boldsymbol{s}, \boldsymbol{a}) - \beta D_{\mathrm{KL}}(\pi(\cdot|\boldsymbol{s})\|\pi_{\mathrm{d}}(\cdot|\boldsymbol{s}))\right] \\
&= \frac{1}{1-\gamma}\mathbb{E}_{\boldsymbol{s}\sim d_\pi, \boldsymbol{a}\sim\pi}\left[\mathcal{R}(\boldsymbol{s}, \boldsymbol{a}) - \beta \log \frac{\pi(\boldsymbol{a}|\boldsymbol{s})}{\pi_{\mathrm{d}}(\boldsymbol{a}|\boldsymbol{s})}\right],
\end{aligned}
\tag{9}
$$

here $\beta > 0$ is an unknown trade-off parameter that is related to fine-tuning steps. $J_{\mathrm{f}}(\pi)$ can be interpreted as penalizing the probability offset of the policy in $(\boldsymbol{s}, \boldsymbol{a})$ over $\pi$ and $\pi_{\mathrm{d}}$. More specifically, it

---

**Algorithm 2** Diffusion Policy Fine-tuning of ExDM

---

**Require:** Environment $\mathcal{M}$ with rewards $\mathcal{R}$, replay buffer $\mathcal{D}$, pre-trained diffusion policy $\pi_d$ parameterized with the score model $\boldsymbol{\epsilon}_\theta$, fine-tuned diffusion policy $\boldsymbol{\epsilon}_\psi$.
 1: **for** update iteration $n = 1, 2, ..., N$ **do**
 2:     Sample $\boldsymbol{s}$-$\boldsymbol{a}$-$\boldsymbol{r}$ pairs $\{(\boldsymbol{s}^m, \boldsymbol{a}^m, \boldsymbol{r}^m)\}_{m=1}^M$ from $\mathcal{D}$.
 3:     Update Q function with IQL and update Guidance $f_{\phi_{n-1}}$ with CEP.
 4:     Optimize $\psi$ by score distillation with Eq. (14).
 5:     **for** interaction step $= 1, 2, ..., S$ **do**
 6:         Interact with $\mathcal{M}$ by $\boldsymbol{\epsilon}_\psi$ and store state-action-reward pairs into $\mathcal{D}$.
 7:     **end for**
 8: **end for**

---

aims to maximize a surrogate reward of the form $\mathcal{R}(\boldsymbol{s}, \boldsymbol{a}) - \beta \log \frac{\pi(\boldsymbol{a}|\boldsymbol{s})}{\pi_d(\boldsymbol{a}|\boldsymbol{s})}$. However, this surrogate reward depends on the policy $\pi$, and we cannot directly apply the classical RL analyses. Inspired by soft RL (Haarnoja et al., 2017) and offline RL (Peng et al., 2019), we define our Q functions as:

$$Q_\pi(\boldsymbol{s}, \boldsymbol{a}) = \mathbb{E}\left[\mathcal{R}(\boldsymbol{s}, \boldsymbol{a}) + \sum_{i=1}^{\infty} \gamma^i \left(\mathcal{R}(\boldsymbol{s}_i, \boldsymbol{a}_i) - \beta \log \frac{\pi(\boldsymbol{a}_i|\boldsymbol{s}_i)}{\pi_d(\boldsymbol{a}_i|\boldsymbol{s}_i)}\right)\right]. \tag{10}$$

Based on this Q function, we can simplify $J_f$ as

$$J_f(\pi) = \mathbb{E}_{\boldsymbol{s} \sim \rho_0, \boldsymbol{a} \sim \pi}\left[Q_\pi(\boldsymbol{s}, \boldsymbol{a}) - \beta D_{\mathrm{KL}}(\pi(\cdot|\boldsymbol{s}) \| \pi_d(\cdot|\boldsymbol{s}))\right]. \tag{11}$$

To optimize $J_f$, ExDM decouples optimizing Q functions and diffusion policies. In detail, we initial $\pi_0 = \pi_d, Q_0 = Q_{\pi_0}$, then for $n = 1, 2, ...$, we set

$$\pi_n(\cdot|\boldsymbol{s}) \triangleq \arg\max_\pi \mathbb{E}_{\boldsymbol{a} \sim \pi}\left[Q_{\pi_{n-1}}(\boldsymbol{s}, \boldsymbol{a}) - \beta D_{\mathrm{KL}}(\pi(\cdot|\boldsymbol{s}) \| \pi_d(\cdot|\boldsymbol{s}))\right] = \frac{\pi_d(\boldsymbol{a}|\boldsymbol{s}) e^{Q_{n-1}(\boldsymbol{s}, \boldsymbol{a})/\beta}}{Z(\boldsymbol{s})}, \tag{12}$$

$$Q_n \triangleq Q_{\pi_n},$$

here $Z(\boldsymbol{s}) = \int \pi_d(\boldsymbol{a}|\boldsymbol{s}) e^{Q_{n-1}(\boldsymbol{s}, \boldsymbol{a})/\beta} d\boldsymbol{a}$. Building on soft RL analysis (Haarnoja et al., 2017; 2018), we show the policy improvement of each iteration and the optimality of the alternating optimization:

**Theorem 4.2** (Optimality of ExDM, Proof in Appendix B.2). *ExDM can achieve policy improvement, i.e., $J_f(\pi_n) \geq J_f(\pi_{n-1})$ for $\forall n \geq 1$. And $\pi_n$ will converge to the optimal policy of $J_f$.*

Compared with offline RL, in which Q functions are related to offline datasets, Q functions here are related to current policies, which introduces extra challenges as Q functions change correspondingly during fine-tuning. Below, we introduce the practical diffusion policy fine-tuning method of ExDM for both updating Q functions and diffusion policies, respectively (pseudo-code in Algorithm 2).

**Q function optimization.** Our principle for updating Q functions is to penalize actions with large log probability ratios between $\pi$ and $\pi_d$. Thus, we apply implicit Q-learning (IQL) (Kostrikov et al., 2022), which leverages expectile regression to penalize out-of-distribution actions (Appendix C.3).

**Diffusion policy distillation.** At each iteration $n$ with Q function $Q_{n-1}$, calculating $\pi_n$ by Eq. (12) is difficult as $Z(s)$ is a complicated integral. However, sampling from $\pi_n$ can be regarded as sampling from $\pi_d$ with energy guidance $Q_{n-1}$, i.e., guided sampling (Janner et al., 2022). Especially, we employ contrastive energy prediction (CEP) (Lu et al., 2023) to sample from $\propto \pi_d e^{Q_{n-1}/\beta}$ and parameterize $f_{\phi_{n-1}}(\boldsymbol{s}, \boldsymbol{a}_t, t)$ to represent timestep $t$'s energy guidance, which can be optimized as:

$$\min_{\phi_{n-1}} \mathbb{E}_{t, \boldsymbol{s}} \mathbb{E}_{\boldsymbol{a}^1, ..., \boldsymbol{a}^K \sim \pi_d(\cdot|\boldsymbol{s})}\left[-\sum_{i=1}^K \frac{e^{Q_{n-1}(\boldsymbol{s}, \boldsymbol{a}^i)/\beta}}{\sum_{j=1}^K e^{Q_{n-1}(\boldsymbol{s}, \boldsymbol{a}^j)/\beta}} \log \frac{f_{\phi_{n-1}}(\boldsymbol{s}, \boldsymbol{a}_t^i, t)}{\sum_{j=1}^K f_{\phi_{n-1}}(\boldsymbol{s}, \boldsymbol{a}_t^j, t)}\right]. \tag{13}$$

Then ExDM fine-tunes diffusion policies by distilling the score of $\pi_n$ parameterized by $\boldsymbol{\epsilon}_\psi(\boldsymbol{a}_t|\boldsymbol{s}, t)$:

$$\min_\psi \mathbb{E}_{\boldsymbol{s}, \boldsymbol{a}, t} \|\boldsymbol{\epsilon}_\psi(\boldsymbol{a}_t|\boldsymbol{s}, t) - \boldsymbol{\epsilon}_\theta(\boldsymbol{a}_t|\boldsymbol{s}, t) - f_{\phi_{n-1}}(\boldsymbol{s}, \boldsymbol{a}_t, t)\|^2. \tag{14}$$

Finally, we can directly sample from $\boldsymbol{\epsilon}_\psi$ to generate action of $\pi_n$ (details are in Appendix C.4).

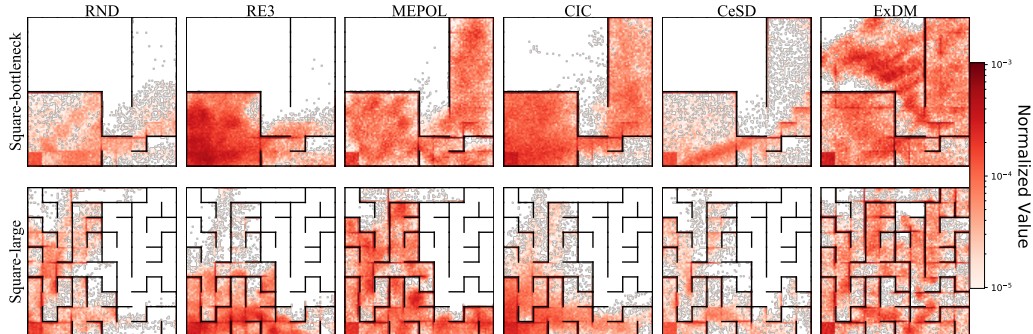

Figure 2: **Heatmap of explored regions** by URL methods in the most complicated mazes.

| Domains | Square-a | Square-b | Square-c | Square-d | Square-tree | Square-bottleneck | Square-large |
|---|---|---|---|---|---|---|---|
| ICM | 0.58 ± 0.04 | 0.53 ± 0.06 | 0.47 ± 0.07 | 0.49 ± 0.06 | 0.49 ± 0.05 | 0.32 ± 0.07 | 0.25 ± 0.04 |
| RND | 0.50 ± 0.14 | 0.39 ± 0.08 | 0.52 ± 0.16 | 0.32 ± 0.05 | 0.28 ± 0.06 | 0.33 ± 0.06 | 0.33 ± 0.08 |
| Disagreement | 0.38 ± 0.10 | 0.30 ± 0.10 | 0.41 ± 0.19 | 0.29 ± 0.11 | 0.32 ± 0.11 | 0.28 ± 0.04 | 0.21 ± 0.06 |
| LBS | 0.32 ± 0.04 | 0.29 ± 0.09 | 0.27 ± 0.05 | 0.25 ± 0.03 | 0.22 ± 0.03 | 0.21 ± 0.02 | 0.19 ± 0.06 |
| RE3 | 0.85 ± 0.09 | 0.72 ± 0.22 | 0.73 ± 0.16 | 0.74 ± 0.01 | 0.73 ± 0.04 | 0.62 ± 0.01 | 0.46 ± 0.03 |
| MEPOL | **0.98 ± 0.03** | **0.99 ± 0.02** | **0.96 ± 0.07** | **0.77 ± 0.01** | **0.89 ± 0.06** | 0.62 ± 0.01 | 0.59 ± 0.04 |
| DIAYN | 0.41 ± 0.06 | 0.44 ± 0.04 | 0.42 ± 0.04 | 0.37 ± 0.03 | 0.38 ± 0.06 | 0.29 ± 0.04 | 0.30 ± 0.04 |
| SMM | 0.47 ± 0.13 | 0.45 ± 0.20 | 0.36 ± 0.08 | 0.28 ± 0.04 | 0.25 ± 0.02 | 0.41 ± 0.13 | 0.34 ± 0.10 |
| LSD | 0.45 ± 0.03 | 0.38 ± 0.05 | 0.36 ± 0.03 | 0.35 ± 0.03 | 0.28 ± 0.03 | 0.34 ± 0.03 | 0.32 ± 0.03 |
| CIC | 0.94 ± 0.02 | **0.98 ± 0.01** | 0.86 ± 0.03 | 0.74 ± 0.01 | **0.89 ± 0.01** | 0.58 ± 0.05 | 0.47 ± 0.01 |
| BeCL | 0.50 ± 0.08 | 0.48 ± 0.11 | 0.42 ± 0.10 | 0.37 ± 0.03 | 0.36 ± 0.06 | 0.29 ± 0.06 | 0.25 ± 0.05 |
| CeSD | 0.70 ± 0.04 | 0.79 ± 0.04 | 0.67 ± 0.06 | 0.46 ± 0.06 | 0.37 ± 0.06 | 0.46 ± 0.03 | 0.40 ± 0.01 |
| ExDM (Ours) | **0.99 ± 0.02** | **0.99 ± 0.01** | **0.98 ± 0.02** | **0.78 ± 0.01** | **0.91 ± 0.01** | **0.75 ± 0.15** | **0.71 ± 0.07** |

Table 1: **State coverage in Maze**. We report the mean and std of 10 seeds for each algorithm.

## 5 EXPERIMENTS

In this section, we present extensive empirical results to mainly address the following questions:

- Can ExDM boost the unsupervised exploration efficiency, especially in complicated mazes with numerous branching paths and decision points? (Sec. 5.2)
- What about the adaptation efficiency of the pre-trained Gaussian policies of ExDM compared to other URL baselines? (Sec. 5.3)
- As for fast fine-tuning pre-trained diffusion policies to downstream tasks, how does the performance of ExDM compare to existing baselines? (Sec. 5.4)

### 5.1 EXPERIMENTAL SETUP

**Maze2d.** We first evaluate the exploration diversity during the unsupervised stage in widely used maze2d environments (Campos et al., 2020; Yang et al., 2023b): Square-a, Square-b, Square-c, Square-d, Square-tree, Square-bottleneck, and Square-large. Observations and actions here belong to $\mathbb{R}^2$. When interacting with mazes, agents will be blocked when they contact walls.

**Continuous Control.** To evaluate the performance of fine-tuning in downstream tasks, we choose 4 continuous control settings in URLB (Laskin et al., 2021): Walker, Quadruped, Jaco, and Hopper. Each domain contains four downstream tasks. More details are in Appendix D.1.

**Baselines.** In **Maze2d** experiments, we take 6 **exploration baselines**: ICM (Pathak et al., 2017), RND (Burda et al., 2018), Disagreement (Pathak et al., 2019), RE3 (Seo et al., 2021), MEPOL (Mutti et al., 2021), and LBS (Mazzaglia et al., 2022); as well as 6 **skill discovery baselines**: DIAYN (Eysenbach et al., 2018), SMM (Lee et al., 2019), LSD (Park et al., 2022), CIC (Laskin et al., 2022), BeCL (Yang et al., 2023b), and CeSD (Bai et al., 2024), which are standard and SOTA. As for fine-tuning in **URLB**, we consider three settings: **(a)** for a fair comparison, we directly utilize

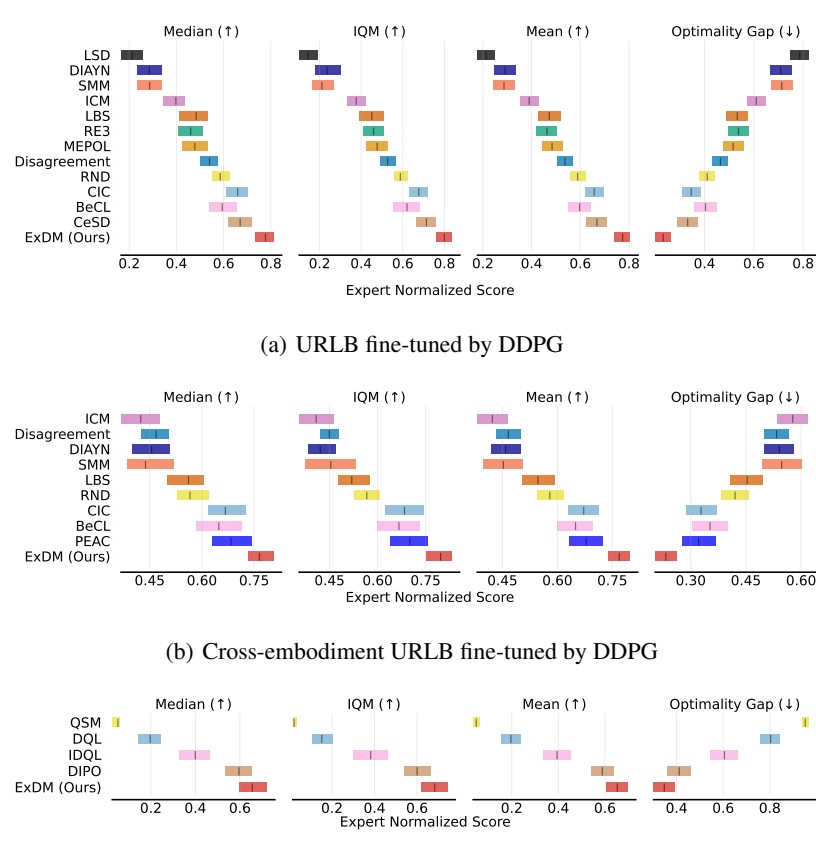

(a) URLB fine-tuned by DDPG

(b) Cross-embodiment URLB fine-tuned by DDPG

(c) URLB for fine-tuning diffusion policies

Figure 3: **Aggregate metrics** (Agarwal et al., 2021) for three settings. Details are in Appendix D.5.

DDPG (Sohl-Dickstein et al., 2015) to fine-tune the pre-trained behavior Gaussian policy in ExDM, compared to existing URL baselines, including ICM, RND, Disagreement, RE3, MEPOL, LBS, DIAYN, SMM, LSD, CIC, BeCL, and CeSD (all baselines fine-tuned by DDPG, the standard RL backbone in URLB, except CeSD fine-tuned by ensembled DDPG; **(b)** We consider complicated cross-embodiment URL (Ying et al., 2024), comparing ExDM with ICM, Disagreement, RND, LBS, DIAYN, SMM, CIC, BeCL, and PEAC (Ying et al., 2024); **(c)** ExDM also fine-tunes pre-trained diffusion policies, compared to diffusion policy fine-tuned baselines, like DQL (Wang et al., 2023), IDQL (Hansen-Estruch et al., 2023), QSM (Psenka et al., 2023), and DIPO (Yang et al., 2023a).

**Metrics.** In Maze2d, we pre-train agents in reward-free environments with **100k** steps and visualize all collected trajectories. Moreover, to quantitatively compare the exploration efficiency, we evaluate the state coverage ratios, which are measured as the proportion of $0.01 \times 0.01$ square bins visited. As for URLB, following standard settings, we pre-train agents in reward-free environments for **2M** steps and fine-tune pre-trained policies to adapt each downstream task within extrinsic rewards for **100K** steps. All settings are run for 10 seeds to mitigate the effectiveness of randomness.

## 5.2 Unsupervised Pre-training for Exploration

In Fig. 2, we visualize the heatmap of collected trajectories during pre-training in complicated Square-bottleneck and Square-large (results of all 7 mazes and 13 baselines are in Appendix D.4). To quantitatively evaluate the exploration efficiency of each algorithm, we further report the state coverages in Table 1 (training curves are in Fig. 8 of Appendix D.4). In both qualitative visualization and quantitative metrics, ExDM outperforms baselines by large margins. Especially, in complicated mazes like Square-bottleneck and Square-large with many different branching points (Fig. 2), all baselines will struggle at some wall corner and cannot explore the entire maze. In contrast, ExDM

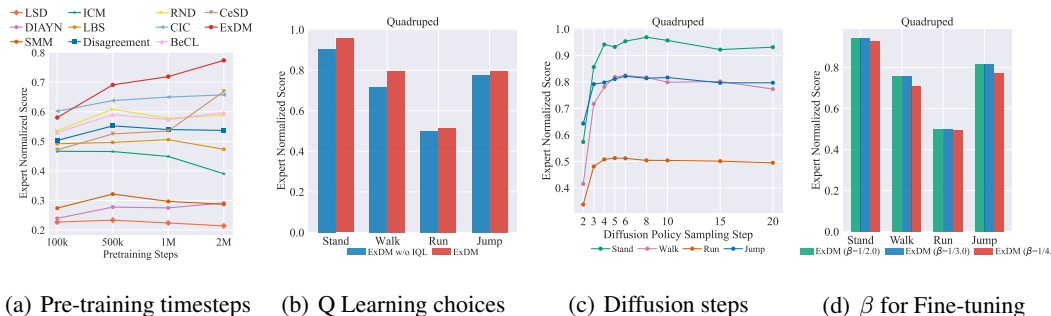

| (a) Pre-training timesteps | (b) Q Learning choices | (c) Diffusion steps | (d) $\beta$ for Fine-tuning |

Figure 4: **Ablation studies**.

successfully explores almost the whole maze, demonstrating that our $\mathcal{R}_{\text{score}}$, leveraging the accurate data estimation ability of diffusion models, can effectively guide agents to explore diverse states.

### 5.3 FINE-TUNING THE GAUSSIAN POLICY TO DOWNSTREAM TASKS

We verify the ability of ExDM to fine-tune downstream tasks in both single-embodiment and cross-embodiment URLB. As existing URL methods directly fine-tune policies with DDPG, for a fair comparison, we also use DDPG to fine-tune pre-trained Gaussian policies $\pi_g$ in ExDM. Following previous settings, we train DDPG agents for each downstream task with 2M steps to obtain the expert return and calculate the expert-normalized score for each algorithm. In Fig. 3(a)-3(b), we compare all methods with four metrics: mean, median, interquartile mean (IQM), and optimality gap (OG), along with stratified bootstrap confidence intervals. ExDM significantly outperforms all existing SOTA methods, demonstrating that introducing diffusion models can lead to more efficient generalization in downstream tasks. Details of each downstream task are in Appendix D.5.

### 5.4 FINE-TUNING THE DIFFUSION POLICY TO DOWNSTREAM TASKS

Moreover, ExDM has pre-trained diffusion policies that can capture the diversity of explored trajectories and adapt to downstream tasks. Consequently, in Fig. 3(c), ExDM substantially outperforms existing diffusion online fine-tuning baselines, demonstrating the efficiency of its alternating optimization. However, diffusion policy fine-tuning performance is still lower than Gaussian policy performance, which may be due to limited interaction timesteps during fine-tuning. It is an interesting future direction to design more efficient diffusion online fine-tuning methods.

### 5.5 ABLATION STUDIES

**Pre-training Steps.** We first do ablation studies on pre-trained steps (100k, 500k, 1M, and 2M) to evaluate fine-tuned performance (100k fine-tuned steps). As shown in Fig. 4(a), ExDM markedly exceeds all baselines from 500k steps, indicating that the diffusion model enhances fine-tuning. Moreover, ExDM substantially improves with increasing pre-training timesteps, showing that the unsupervised exploration benefits downstream tasks. Additional results are in Appendix D.6.

**Q function optimization.** We conduct ablation studies to evaluate the impact of Q learning methods during fine-tuning. In detail, we introduce ExDM w/o IQL, which utilizes In-support Softmax Q-Learning (Lu et al., 2023) for optimizing Q functions. Fig. 4(b) demonstrates that ExDM consistently outperforms ExDM w/o IQL, verifying the efficiency of IQL in diffusion policy fine-tuning.

**Sampling steps of diffusion policies.** During fine-tuning diffusion policies, ExDM requires sampling actions from diffusion policies for both trajectory generation and final evaluation. We adopt DPM-Solver (Lu et al., 2022) to accelerate sampling. For trajectory collection, we set the diffusion step to 15, following previous works (Lu et al., 2023). Then we conduct ablation studies between the diffusion sampling steps used during inference and the fine-tuned performance. Fig. 4(c) shows that performance improves as diffusion steps increase and gradually stabilizes when the step exceeds 5.

**Ablation study on $\beta$.** We consider the objective $J_{\mathrm{f}}$ with the behavior regularization term because the fine-tuning step is limited (more analyses are in Appendix C.2). The parameter $\beta$ implicitly relies on the fine-tuning steps. If the fine-tuning steps are infinite, the optimal $\beta$ should be 0, and $J_{\mathrm{f}}$ degrades into $J$. We set $\beta = 1/3.0$ in experiments (following previous work (Lu et al., 2023)) and do ablations with different $\beta$ in Fig. 4(d), showing that ExDM performs relatively stably of $\beta$.

**Time cost of ExDM.** One of the major concerns for diffusion models is their time cost due to multi-step sampling. This problem may be more severe in online RL, as each collected trajectory requires sampling from diffusion policies. To address it, ExDM decouples modeling from acting, i.e., utilizing Gaussian behavior policies $\pi_{\mathrm{g}}$ for sampling. Thus, ExDM exhibits high training efficiency for both timesteps and training time. For example, in Square-large, RND achieves the state coverage of 0.33 with 100k timesteps and 1000s of time. ExDM can achieve 0.71 state coverage within 100k timesteps, and achieve 0.4 state coverage with the same time cost (1000s) and only 16.6k timesteps.

## 6    CONCLUSION

Unsupervised exploration is one of the major problems in RL for improving task generalization, as it relies on accurate intrinsic rewards to guide exploration of unseen regions. In this work, we address the challenge of limited policy expressivity in previous exploration methods by leveraging the powerful expressive ability of diffusion policies. In detail, our Exploratory Diffusion Model (ExDM) improves exploration efficiency during pre-training while generating policies with high behavioral diversity. We also provide a theoretical analysis of diffusion policy fine-tuning, along with practical alternating optimization methods. Experiments in various settings demonstrate that ExDM can effectively benefit both pre-training exploration and fine-tuning performance. We hope this work can inspire further research in developing high-fidelity generative models to improve unsupervised exploration, particularly in large-scale pre-trained agents or real-world control applications.

## ETHICS STATEMENT

Designing generalizable agents for varying tasks is one of the major concerns in reinforcement learning. This work focuses on utilizing diffusion policies for exploration and proposes a novel algorithm ExDM. One of the potential negative impacts is that algorithms mainly use deep neural networks, which lack interoperability and may face robustness issues. There are no serious ethical issues, as this is basic research.

## REPRODUCIBILITY STATEMENT

To ensure that our work is reproducible, we have submitted the source code in https://github.com/yingchengyang/ExDM. We also provide the pseudo-code of ExDM in Algorithm 1-2 and implementation details of ExDM, including hyper-parameters in Appendix D. Moreover, for all theoretical results, we have provided all details and proofs in Appendix B.

## ACKNOWLEDGEMENT

This project is supported by Fundamental and Interdisciplinary Disciplines Breakthrough Plan of the Ministry of Education of China (No. JYB2025XDXM101), NSF of China Projects (Nos. U25B6003, 62550004, 92370124, 92248303); Beijing Natural Science Foundation L247011; the High Performance Computing Center, Tsinghua University. J.Z was also supported by the XPlorer Prize.

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

# A EXTENDED RELATED WORK

## A.1 META RL ANE UNSUPERVISED RL

Improving the generalization of RL agents is a long-term challenge. There are several settings aiming at pre-training agents and then fine-tuning the pre-trained agent to fast adapt to various tasks, like unsupervised RL and meta RL. Unsupervised RL, as we have discussed above, pre-trains agents in **reward-free environments**, i.e., the environment without task rewards. Thus, we will design intrinsic rewards to pre-train agents to fully explore the environment. And the pre-trained agent needs to fast adapt to downstream tasks that it has not seen during the pre-training stage. Meta RL pre-trains agents in **several sampled tasks from a task distribution** $\mathcal{T}$. And the pre-trained agent needs to fast adapt to test tasks that are sampled from $\mathcal{T}$ but are not seen during the pre-training stage. There are two major types of methods in Meta RL: gradient-based (Finn et al., 2017; Flennerhag et al., 2021; Liu et al., 2022) and context-based methods (Duan et al., 2016; Rakelly et al., 2019; Zintgraf et al., 2019). Besides model-free methods, there are also some methods that apply learned dynamic models to boost the sample efficiency of meta RL (Nagabandi et al., 2018; Clavera et al., 2018; Hiraoka et al., 2021). Consequently, the major difference between unsupervised RL and meta RL is that, during the pre-training stage, there are no task rewards in the former setting, but the agent can access several task rewards sampled from the same distribution of the test tasks in the latter setting.

## A.2 ONLINE FINE-TUNING DIFFUSION POLICIES

Although diffusion policies have been proven to be expressive models for capturing diverse offline datasets (Wang et al., 2023; Chen et al., 2023), online fine-tuning diffusion policies (Psenka et al., 2023; Li et al., 2024; Ren et al., 2024; Mark et al., 2024; Celik et al., 2025; Ma et al., 2025; Ishfaq et al., 2025) still face many challenges and are a key issue of concern for the community. For example, the instability caused by multi-step sampling and the lack of closed-form probability calculation are significant concerns. Many recent works hope to address these challenges and improve the efficiency of online training diffusion policies; they mainly consider two settings: online training diffusion policies from scratch with enough timesteps (Psenka et al., 2023; Yang et al., 2023a) or online fine-tuning diffusion policies pre-trained from an offline dataset sampled from the same task (Hansen-Estruch et al., 2023; Ma et al., 2025). Furthermore, there are several potential directions for improving fine-tuning efficiency, including: more energy-guided sampling techniques Liu et al. (2024), more efficient diffusion sampling techniques (e.g., distillation, deterministic sampling) (Wang et al., 2024b), and so on.

## B  THEORETICAL ANALYSES

### B.1  THE DETAILS AND PROOF OF THEOREM 4.1

In this part, we narrate Theorem 4.1 in detail as well as provide its proof.

Assume that $\mathcal{S}, \mathcal{A}$ are discrete spaces and $|\mathcal{S}| = S, |\mathcal{A}| = A$. For any policy $\pi : \mathcal{S} \to \Delta(\mathcal{A})$, we define the discount state distribution of $\pi$ as $d_\pi(\boldsymbol{s}) = (1-\gamma) \sum_{t=0}^\infty [\gamma^t \mathcal{P}(\boldsymbol{s}^t = \boldsymbol{s})]$. Naturally, $d_\pi(\cdot)$ is a distribution of the state space $\mathcal{S}$. Specially, when $\mathcal{S}$ is discrete, following previous work (Eysenbach et al., 2021; Ying et al., 2024), we can regard $d_\pi$ as a point of the probability simplex in $\mathbb{R}^S$, i.e., $d_\pi \in H \triangleq \{(x_i)_{i=1}^S | \sum_{i=1}^S x_i = 1, 0 \le x_i \le 1\}$ (for example, the light blue plane in the left part of Fig. 1). Moreover, we consider $D \triangleq \{d_\pi \in H | \forall \pi\} \subseteq H$ representing all feasible state distribution.

It is natural that there are $M \triangleq A^S$ different deterministic policies. Previous work (Eysenbach et al., 2021) has proven that $D$ is a convex polytope, of which the vertices are the state distributions of the deterministic policies (for example, the green points of Fig. 1). Consequently, for any downstream task represented by some extrinsic reward function $\mathcal{R}$, the optimal policy is one the vertices of $D$, i.e., some deterministic policy.

Differently, the unsupervised exploration in URL aims to maximize the entropy of the policy, i.e., we hope to optimize $\hat{\pi} = \arg\max_\pi \mathcal{H}(d_\pi(\cdot))$. As $\mathcal{H}(d_\pi(\cdot)) = \int_{\boldsymbol{s} \sim d_\pi(\cdot)} [-\log d_\pi(\boldsymbol{s})] d\boldsymbol{s}$, this problem can be regard as maximizing a surrogate reward $-\log d_\pi(\boldsymbol{s})$. However, the surrogate reward is related to the current policy $\pi$, thus the analyses of standard RL may not hold. Actually, if we consider all distribution over $\mathcal{S}$, i.e., all points in $H$, it is well known that the distribution with the maximal distribution is the center of $H$, i.e., $O = (1/S, 1/S, ..., 1/S) \in H$ (for example, the red point of Fig. 1). Although $O \in D$ may not hold, we hope to claim that the optimal policy $\hat{\pi}$ with the maximal state distribution entropy may not be deterministic, and its state distribution is $O$, i.e., $O \in D$, with high probability.

We consider the distribution in which the state distributions of the $M$ deterministic policies are i.i.d., and all follow the uniform distribution on $H$. Therefore, our problem can be transformed into: *there are $M$ i.i.d. points uniformly sampled from $H$, and the convex polytope formed by these $M$ points is $D \subseteq H$, then calculating the probability of the event $O \in D$.*

Based on the results in geometric probability (Eq. (13) of paper (Baddeley et al., 2007), extending the Wendel Theorem in (Wendel, 1962)), we have

$$P(O \in D) \ge 1 - \sum_{k=0}^{S-1} C_M^k \left(\frac{u(O)}{2}\right)^k \left(1 - \frac{u(O)}{2}\right)^{M-k}, \tag{15}$$

here $u(O) = \mathrm{vol}((2O - H) \cap H)/\mathrm{vol}(H)$. Below we first estimate $u(O)$, which is obviously belong to $[0, 1]$. We have

$$
\begin{aligned}
\mathrm{vol}(H) &= \sqrt{S} \int_{x_1+x_2+...+x_S=1, x_i \ge 0} dx_1 dx_2 ... dx_S \\
&= \sqrt{S} \int_0^1 dx_1 \int_0^{1-x_1} dx_2 ... \int_0^{1-x_1-x_2-...-x_{S-2}} dx_{S-1} \\
&= \sqrt{S} \int_0^1 dx_1 \int_0^{1-x_1} dx_2 ... \int_0^{1-x_1-x_2-...-x_{S-3}} (1 - x_1 - ... - x_{S-2}) dx_{S-2} \\
&= \sqrt{S} \int_0^1 dx_1 \int_0^{1-x_1} dx_2 ... \int_0^{1-x_1-x_2-...-x_{S-4}} \frac{(1 - x_1 - ... - x_{S-3})^2}{2} dx_{S-3} \\
&= ... \\
&= \sqrt{S} \int_0^1 \frac{(1-x_1)^{S-2}}{(S-2)!} dx_1 \\
&= \frac{\sqrt{S}}{(S-1)!}.
\end{aligned}
\tag{16}
$$

Then we estimate $\text{vol}((2O - H) \cap H)$. $H$ is a convex polytope surrounded by $S$ points $A_1 = (1, 0, 0, ..., 0), ..., A_S = (0, 0, ..., 0, 1)$. As $O = (1/S, 1/S, ..., 1/S)$, $2O - H$ is a convex polytope surrounded by $S$ points $B_1 = (2/S - 1, 1/S, 1/S, ..., 1/s), ..., B_S = (1/S, 1/S, ..., 1/S, 2/S - 1)$. It is difficult to directly calculate the volume of $(2O - H) \cap H$ (although we can calculate it by the inclusion-exclusion principle). We can provide a lower bound of $(2O - H) \cap H$. Consider a convex polytope $C$ surrounded by $S$ points $C_1 = (0, 1/(S-1), 1/(S-1), ..., 1/(S-1)), ..., C_S = (1/(S-1), 1/(S-1), ..., 1/(S-1), 0)$. It is easy to show that $C \subseteq (2O - H) \cap H$ and $C \sim H$. As the side length of $H$ and $C$ is $\sqrt{\frac{2}{S}}$ and $\sqrt{\frac{2}{S}}\frac{1}{S-1}$. Thus we have $\text{vol}((2O - H) \cap H) \geq \text{vol}(C) = \frac{1}{(S-1)^S}\frac{\sqrt{S}}{(S-1)!}$ and $u(O) = \text{vol}((2O - H) \cap H)/\text{vol}(H) \geq \frac{1}{(S-1)^S}$. Assume that $M \geq S - 2 + (S-1)\frac{2-u(O)}{u(O)}$, then for $\forall 0 \leq k \leq S - 2$, we have

$$
\begin{aligned}
& C_M^k \left(\frac{u(O)}{2}\right)^k \left(1 - \frac{u(O)}{2}\right)^{M-k} \\
=& \frac{k+1}{M-k}\frac{2-u(O)}{u(O)} C_M^{k+1} \left(\frac{u(O)}{2}\right)^{k+1} \left(1 - \frac{u(O)}{2}\right)^{M-k-1} \\
\leq& \frac{S-2+1}{M-S+2}\frac{2-u(O)}{u(O)} C_M^{k+1} \left(\frac{u(O)}{2}\right)^{k+1} \left(1 - \frac{u(O)}{2}\right)^{M-k-1} \\
\leq& \frac{S-1}{(S-1)\frac{2-u(O)}{u(O)}}\frac{2-u(O)}{u(O)} C_M^{k+1} \left(\frac{u(O)}{2}\right)^{k+1} \left(1 - \frac{u(O)}{2}\right)^{M-k-1} \\
=& C_M^{k+1} \left(\frac{u(O)}{2}\right)^{k+1} \left(1 - \frac{u(O)}{2}\right)^{M-k-1}.
\end{aligned}
\tag{17}
$$

Consequently, we set $v(S) = 1 - \frac{u(O)}{2} < 1$ and have

$$
\begin{aligned}
P(O \in D) \geq& 1 - \sum_{k=0}^{S-1} C_M^k \left(\frac{u(O)}{2}\right)^k \left(1 - \frac{u(O)}{2}\right)^{M-k} \\
\geq& 1 - C_M^{S-1} S \left(\frac{u(O)}{2}\right)^{S-1} \left(1 - \frac{u(O)}{2}\right)^{M-S+1} \\
=& 1 - C_M^{S-1} S \left(\frac{u(O)}{2-u(O)}\right)^{S-1} \left(1 - \frac{u(O)}{2}\right)^{M} \\
\geq& 1 - C_M^{S-1} S v(S)^M \\
=& 1 - \frac{M \times (M-1) \times ... \times (M-S+2)}{1 \times 2 \times ... \times (S-1)} S v(S)^M \\
\geq& 1 - M \times (M-1) \times ... \times (M-S+2) \times S v(S)^M \\
\geq& 1 - M^S v(S)^M.
\end{aligned}
\tag{18}
$$

If we fix $S$, with the increasing of $A$, $M^S v(S)^M$ will fast converge to 0 as $0 < v(S) < 1$. Moreover, the right part of the first line of Eq. 18 is always larger than 0 when $M \geq S$ (the sum of these combination numbers is always less than 1 when $M > S$). Thus, we have proven Theorem 4.1.

Below, we will discuss the situation when the state space and action space are continuous. As the optimal state distribution with the maximal state entropy is somewhat uniform, in continuous environments, there may still not be simple unimodal distributions. For example, assume a one-step MDP, we only take one action from $[0, 1]$, and the state is the same as the action, thus the optimal policy should be the uniform distribution over $[0, 1]$ rather than simple unimodal distributions.

## B.2 THE PROOF OF THEOREM 4.2

Below, we first analyze our fine-tuning objective Eq. 9 and then prove Theorem 4.2.

Assuming $\rho_0$ is the original state distribution of the MDP $\mathcal{M}$, we have

$$
\begin{aligned}
J_{\mathrm{f}}(\pi) \triangleq & J(\pi) - \frac{\beta}{1-\gamma} \mathbb{E}_{\boldsymbol{s} \sim d_\pi}\left[D_{\mathrm{KL}}(\pi(\cdot|\boldsymbol{s})\|\pi_{\mathrm{d}}(\cdot|\boldsymbol{s}))\right] \\
= & \frac{1}{1-\gamma} \mathbb{E}_{\boldsymbol{s} \sim d_\pi, \boldsymbol{a} \sim \pi(\cdot|\boldsymbol{s})}\left[\mathcal{R}(\boldsymbol{s}, \boldsymbol{a}) - \beta D_{\mathrm{KL}}(\pi(\cdot|\boldsymbol{s})\|\pi_{\mathrm{d}}(\cdot|\boldsymbol{s}))\right] \\
= & \mathbb{E}_{\boldsymbol{s} \sim \rho_0, \boldsymbol{a} \sim \pi(\cdot|\boldsymbol{s})}\left[\sum_{i=0}^{\infty} \gamma^i\left(\mathcal{R}(\boldsymbol{s}_i, \boldsymbol{a}_i) - \beta D_{\mathrm{KL}}(\pi(\cdot|\boldsymbol{s}_i)\|\pi_{\mathrm{d}}(\cdot|\boldsymbol{s}_i))\right)\bigg| \boldsymbol{s}_0 = \boldsymbol{s}, \boldsymbol{a}_0 = \boldsymbol{a}\right] \\
= & \mathbb{E}_{\boldsymbol{s} \sim \rho_0, \boldsymbol{a} \sim \pi(\cdot|\boldsymbol{s})}\bigg[\mathcal{R}(\boldsymbol{s}, \boldsymbol{a}) - \beta D_{\mathrm{KL}}(\pi(\cdot|\boldsymbol{s})\|\pi_{\mathrm{d}}(\cdot|\boldsymbol{s})) \\
& \qquad\qquad + \sum_{i=1}^{\infty} \gamma^i\left(\mathcal{R}(\boldsymbol{s}_i, \boldsymbol{a}_i) - \beta D_{\mathrm{KL}}(\pi(\cdot|\boldsymbol{s}_i)\|\pi_{\mathrm{d}}(\cdot|\boldsymbol{s}_i))\right)\bigg] \\
= & \mathbb{E}_{\boldsymbol{s} \sim \rho_0, \boldsymbol{a} \sim \pi(\cdot|\boldsymbol{s})}\left[Q_\pi(\boldsymbol{s}, \boldsymbol{a}) - \beta D_{\mathrm{KL}}(\pi(\cdot|\boldsymbol{s})\|\pi_{\mathrm{d}}(\cdot|\boldsymbol{s}))\right],
\end{aligned}
\tag{19}
$$

here we set

$$
\begin{aligned}
Q_\pi(\boldsymbol{s}, \boldsymbol{a}) &= \mathbb{E}\left[\mathcal{R}(\boldsymbol{s}, \boldsymbol{a}) + \sum_{i=1}^{\infty} \gamma^i\left(\mathcal{R}(\boldsymbol{s}_i, \boldsymbol{a}_i) - \beta \log \frac{\pi(\boldsymbol{a}_i|\boldsymbol{s}_i)}{\pi_{\mathrm{d}}(\boldsymbol{a}_i|\boldsymbol{s}_i)}\right)\right] \\
&= \mathbb{E}\left[\mathcal{R}(\boldsymbol{s}, \boldsymbol{a}) + \sum_{i=1}^{\infty} \gamma^i\left(\mathcal{R}(\boldsymbol{s}_i, \boldsymbol{a}_i) - \beta D_{\mathrm{KL}}(\pi(\cdot|\boldsymbol{s}_i)\|\pi_{\mathrm{d}}(\cdot|\boldsymbol{s}_i))\right)\right].
\end{aligned}
\tag{20}
$$

As discussed in Sec. 4.2, ExDM applies the following alternative optimization method:

$$
\begin{aligned}
\pi_n(\cdot|\boldsymbol{s}) &= \arg\max_\pi \mathbb{E}_{\boldsymbol{a} \sim \pi(\cdot|\boldsymbol{s})}[Q_{\pi_{n-1}}(\boldsymbol{s}, \boldsymbol{a}) - \beta D_{\mathrm{KL}}(\pi(\cdot|\boldsymbol{s})\|\pi_{\mathrm{d}}(\cdot|\boldsymbol{s}))], \\
Q_n &= Q_{\pi_n},
\end{aligned}
\tag{21}
$$

Now we prove that $\pi_n(\boldsymbol{a}|\boldsymbol{s}) = \frac{1}{Z(\boldsymbol{s})}\pi_{\mathrm{d}}(\boldsymbol{a}|\boldsymbol{s})e^{Q_{n-1}(\boldsymbol{s},\boldsymbol{a})/\beta}$. More generally, we define

$$
F(\pi, \pi', \boldsymbol{s}) = \mathbb{E}_{\boldsymbol{a} \sim \pi(\cdot|\boldsymbol{s})}\left[Q_{\pi'}(\boldsymbol{s}, \boldsymbol{a}) - \beta D_{\mathrm{KL}}(\pi(\cdot|\boldsymbol{s})\|\pi_{\mathrm{d}}(\cdot|\boldsymbol{s}))\right].
\tag{22}
$$

Using the calculus of variations, we can calculate the optimal point $\pi^*$ of $F$ satisfying that

$$
Q_{\pi'}(\boldsymbol{s}, \boldsymbol{a}) = \beta \log \frac{\pi^*(\boldsymbol{a}|\boldsymbol{s})}{\pi_{\mathrm{d}}(\boldsymbol{a}|\boldsymbol{s})} + b\beta,
\tag{23}
$$

here $b$ is a constant not related to $\pi^*$, and we have $\pi^*(\boldsymbol{a}|\boldsymbol{s}) = \pi_{\mathrm{d}}(\boldsymbol{a}|\boldsymbol{s})e^{\frac{Q_{\pi'}(\boldsymbol{s},\boldsymbol{a})}{\beta}-b}$. As $\int \pi^*(\boldsymbol{a}|\boldsymbol{s})d\boldsymbol{a} = 1$, we can calculate that

$$
b = \log \int \pi_{\mathrm{d}}(\boldsymbol{a}|\boldsymbol{s})e^{\frac{Q_{\pi'}(\boldsymbol{s},\boldsymbol{a})}{\beta}}d\boldsymbol{a}, \quad \pi^*(\boldsymbol{a}|\boldsymbol{s}) = \frac{\pi_{\mathrm{d}}(\boldsymbol{a}|\boldsymbol{s})e^{\frac{Q_{\pi'}(\boldsymbol{s},\boldsymbol{a})}{\beta}-b}}{\int \pi_{\mathrm{d}}(\boldsymbol{a}|\boldsymbol{s})e^{\frac{Q_{\pi'}(\boldsymbol{s},\boldsymbol{a})}{\beta}}d\boldsymbol{a}}.
\tag{24}
$$

i.e., we have $\arg\max_\pi F(\pi, \pi', \boldsymbol{s}) \propto \pi_{\mathrm{d}}(\cdot|\boldsymbol{s})e^{Q_{\pi'}(\boldsymbol{s},\cdot)/\beta}$ and thus $\pi_n(\boldsymbol{a}|\boldsymbol{s}) = \frac{1}{Z(\boldsymbol{s})}\pi_{\mathrm{d}}(\boldsymbol{a}|\boldsymbol{s})e^{Q_{n-1}(\boldsymbol{s},\boldsymbol{a})/\beta}$.

Below we will prove Theorem 4.2.

*Proof.* Based on the definition of $F$, we have $J_{\mathrm{f}}(\pi) = \mathbb{E}_{\boldsymbol{s} \sim \rho_0} F(\pi, \pi, \boldsymbol{s})$. Thus we require to prove $\mathbb{E}_{\boldsymbol{s} \sim \rho_0} F(\pi_n, \pi_n, \boldsymbol{s}) \geq \mathbb{E}_{\boldsymbol{s} \sim \rho_0} F(\pi_{n-1}, \pi_{n-1}, \boldsymbol{s})$. As we have discussed above,

$$
\begin{aligned}
\pi_n(\cdot|\boldsymbol{s}) &= \arg\max_\pi F(\pi, \pi_{n-1}, \boldsymbol{s}) = \frac{1}{Z(\boldsymbol{s})}\pi_{\mathrm{d}}(\boldsymbol{a}|\boldsymbol{s})e^{Q_{\pi_{n-1}}(\boldsymbol{s},\boldsymbol{a})/\beta}. \\
F(\pi_n, \pi_{n-1}, \boldsymbol{s}) &\geq F(\pi_{n-1}, \pi_{n-1}, \boldsymbol{s}).
\end{aligned}
\tag{25}
$$

In other words, we have proven that $\mathbb{E}_{\boldsymbol{s} \sim \rho_0} F(\pi_n, \pi_{n-1}, \boldsymbol{s}) \geq \mathbb{E}_{\boldsymbol{s} \sim \rho_0} F(\pi_{n-1}, \pi_{n-1}, \boldsymbol{s})$. Moreover, we have

$$
\begin{aligned}
Q_{\pi_{n-1}}(\boldsymbol{s}, \boldsymbol{a}) =& \mathcal{R}(\boldsymbol{s}, \boldsymbol{a}) + \mathbb{E}\left[\sum_{i=1}^{\infty} \gamma^i \left(\mathcal{R}(\boldsymbol{s}_i, \boldsymbol{a}_i) - \beta D_{\mathrm{KL}}(\pi_{n-1}(\cdot|\boldsymbol{s}_i)\|\pi_{\mathrm{d}}(\cdot|\boldsymbol{s}_i))\right) \bigg| \boldsymbol{s}_0 = \boldsymbol{s}, \boldsymbol{a}_0 = \boldsymbol{a}\right] \\
=& \mathcal{R}(\boldsymbol{s}, \boldsymbol{a}) - \beta\gamma \mathbb{E}_{\boldsymbol{s}_1}\left(D_{\mathrm{KL}}(\pi_{n-1}(\cdot|\boldsymbol{s}_1)\|\pi_{\mathrm{d}}(\cdot|\boldsymbol{s}_1))\right) + \gamma \mathbb{E}_{\boldsymbol{s}_1, \boldsymbol{a}_1}\left[Q_{\pi_{n-1}}(\boldsymbol{s}_1, \boldsymbol{a}_1)\right] \\
=& \mathcal{R}(\boldsymbol{s}, \boldsymbol{a}) + \gamma \mathbb{E}_{\boldsymbol{s}_1} F(\pi_{n-1}, \pi_{n-1}, \boldsymbol{s}).
\end{aligned}
\tag{26}
$$

Thus

$$
\begin{aligned}
& Q_{\pi_n}(\boldsymbol{s}, \boldsymbol{a}) - Q_{\pi_{n-1}}(\boldsymbol{s}, \boldsymbol{a}) \\
=& \gamma \mathbb{E}_{\boldsymbol{s}_1}\left[F(\pi_n, \pi_n, \boldsymbol{s}_1) - F(\pi_{n-1}, \pi_{n-1}, \boldsymbol{s}_1)\right] \geq \gamma \mathbb{E}_{\boldsymbol{s}_1}\left[F(\pi_n, \pi_n, \boldsymbol{s}_1) - F(\pi_n, \pi_{n-1}, \boldsymbol{s}_1)\right] \\
=& \gamma \mathbb{E}_{\boldsymbol{s}_1} \mathbb{E}_{\boldsymbol{a}_1 \sim \pi_n}[Q_{\pi_n}(\boldsymbol{s}_1, \boldsymbol{a}_1) - \beta D_{\mathrm{KL}}(\pi_n(\cdot|\boldsymbol{s}_1)\|\pi_{\mathrm{d}}(\cdot|\boldsymbol{s}_1) \\
& \qquad\qquad - Q_{\pi_{n-1}}(\boldsymbol{s}_1, \boldsymbol{a}_1) + \beta D_{\mathrm{KL}}(\pi_n(\cdot|\boldsymbol{s}_1)\|\pi_{\mathrm{d}}(\cdot|\boldsymbol{s}_1))] \\
=& \gamma \mathbb{E}_{\boldsymbol{s}_1} \mathbb{E}_{\boldsymbol{a}_1 \sim \pi_n}\left[Q_{\pi_n}(\boldsymbol{s}_1, \boldsymbol{a}_1) - Q_{\pi_{n-1}}(\boldsymbol{s}_1, \boldsymbol{a}_1)\right].
\end{aligned}
\tag{27}
$$

Given the property of $d_\pi$ that $d_\pi(\boldsymbol{s}) - (1-\gamma)\rho_0(\boldsymbol{s}) = \gamma \sum_{\boldsymbol{s}'} d_\pi(\boldsymbol{s}') \sum_{\boldsymbol{a}} \pi(\boldsymbol{a}|\boldsymbol{s}')\mathcal{P}(\boldsymbol{s}|\boldsymbol{s}', \boldsymbol{a})$ (Ying et al., 2022), we have

$$
\begin{aligned}
& \mathbb{E}_{\boldsymbol{s} \sim d_{\pi_n}, \boldsymbol{a} \sim \pi_n(\cdot|\boldsymbol{s})}\left[Q_{\pi_n}(\boldsymbol{s}, \boldsymbol{a}) - Q_{\pi_{n-1}}(\boldsymbol{s}, \boldsymbol{a})\right] \\
\geq& \gamma \mathbb{E}_{\boldsymbol{s} \sim d_{\pi_n}, \boldsymbol{a} \sim \pi_n(\cdot|\boldsymbol{s})} \mathbb{E}_{\boldsymbol{s}_1} \mathbb{E}_{\boldsymbol{a}_1 \sim \pi_n}\left[Q_{\pi_n}(\boldsymbol{s}_1, \boldsymbol{a}_1) - Q_{\pi_{n-1}}(\boldsymbol{s}_1, \boldsymbol{a}_1)\right] \\
=& \int \left(d_{\pi_n}(\boldsymbol{s}_1) - (1-\gamma)\rho_0(\boldsymbol{s}_1)\right) \mathbb{E}_{\boldsymbol{a}_1 \sim \pi_n}\left[Q_{\pi_n}(\boldsymbol{s}_1, \boldsymbol{a}_1) - Q_{\pi_{n-1}}(\boldsymbol{s}_1, \boldsymbol{a}_1)\right] d\boldsymbol{s}_1 \\
=& \mathbb{E}_{\boldsymbol{s}_1 \sim d_{\pi_n}, \boldsymbol{a}_1 \sim \pi_n(\cdot|\boldsymbol{s}_1)}\left[Q_{\pi_n}(\boldsymbol{s}_1, \boldsymbol{a}_1) - Q_{\pi_{n-1}}(\boldsymbol{s}_1, \boldsymbol{a}_1)\right] \\
& -(1-\gamma)\mathbb{E}_{\boldsymbol{s}_1 \sim \rho_0, \boldsymbol{a}_1 \sim \pi_n(\cdot|\boldsymbol{s}_1)}\left[Q_{\pi_n}(\boldsymbol{s}_1, \boldsymbol{a}_1) - Q_{\pi_{n-1}}(\boldsymbol{s}_1, \boldsymbol{a}_1)\right].
\end{aligned}
\tag{28}
$$

Consequently,

$$
\begin{aligned}
& \mathbb{E}_{\boldsymbol{s} \sim \rho_0} F(\pi_n, \pi_{n-1}, \boldsymbol{s}) - \mathbb{E}_{\boldsymbol{s} \sim \rho_0} F(\pi_{n-1}, \pi_{n-1}, \boldsymbol{s}) \\
=& \mathbb{E}_{\boldsymbol{s}_1 \sim \rho_0, \boldsymbol{a}_1 \sim \pi_n(\cdot|\boldsymbol{s}_1)}\left[Q_{\pi_n}(\boldsymbol{s}_1, \boldsymbol{a}_1) - Q_{\pi_{n-1}}(\boldsymbol{s}_1, \boldsymbol{a}_1)\right] \geq 0
\end{aligned}
\tag{29}
$$

Finally, we have

$$
J_{\mathrm{f}}(\pi_n) = \mathbb{E}_{\boldsymbol{s} \sim \rho_0} F(\pi_n, \pi_n, \boldsymbol{s}) \geq \mathbb{E}_{\boldsymbol{s} \sim \rho_0} F(\pi_n, \pi_{n-1}, \boldsymbol{s}) \geq \mathbb{E}_{\boldsymbol{s} \sim \rho_0} F(\pi_{n-1}, \pi_{n-1}, \boldsymbol{s}) = J_{\mathrm{f}}(\pi_{n-1}).
\tag{30}
$$

Thus, our policy iteration can improve the performance. Moreover, under some regularity conditions (following Haarnoja et al. (2017; 2018), for example, considering the policy mapping $\tau : \pi \to \arg\max_{\pi'} F(\pi, \pi', \boldsymbol{s})$ and assume that $\tau$ is a compressed mapping with a fixed point), $\pi_n$ converges to $\pi_\infty$, which is the fixed point with the maximal $J_{\mathrm{f}}$. Since non-optimal policies can be improved by our iteration, the converged policy $\pi_\infty$ is optimal for $J_{\mathrm{f}}$. $\qquad\square$

## C  ANALYSES AND DETAILS OF EXDM

Below, we discuss more analyses and details about ExDM, including analyses of $\mathcal{R}_{\text{score}}$, analyses of the objective $J_{\text{f}}$, details of the Q function optimization, and details of diffusion policy optimization of ExDM during the fine-tuning stage.

### C.1  ANALYSES OF THE SCORE INTRINSIC REWARD $\mathcal{R}_{\text{score}}$

It is an important result stemming from diffusion models that the KL divergence between the original distribution and the diffusion model estimated distribution. In detail,

$$
\begin{aligned}
L_{\text{VLB}} =& \mathbb{E}_q \log \frac{q(\boldsymbol{s}_{1:T}|\boldsymbol{s}_0)}{p(\boldsymbol{s}_{0:T})} = -\log p(\boldsymbol{s}_0)\mathbb{E}_q \log \frac{q(\boldsymbol{s}_{1:T}|\boldsymbol{s}_0)}{p(\boldsymbol{s}_{0:T})/p(\boldsymbol{s}_0)} \\
=& -\log p(\boldsymbol{s}_0)\mathbb{E}_q \log \frac{q(\boldsymbol{s}_{1:T}|\boldsymbol{s}_0)}{p(\boldsymbol{s}_{0:T}|\boldsymbol{s}_0)} = -\log p(\boldsymbol{s}_0) + D_{\text{KL}}(p(\boldsymbol{s}_{1:T}|\boldsymbol{s}_0)\|q(\boldsymbol{s}_{1:T}|\boldsymbol{s}_0)),
\end{aligned}
\tag{31}
$$

i.e., $\log p(\boldsymbol{s}_0) + L_{\text{VLB}} = D_{\text{KL}}(p(\boldsymbol{s}_{1:T}|\boldsymbol{s}_0)\|q(\boldsymbol{s}_{1:T}|\boldsymbol{s}_0))$. Consequently, the gap between $L_{\text{VLB}}$ and $-\log p(\boldsymbol{s}_0)$ is the KL divergence between the original distribution and the diffusion model estimated distribution, which will be controlled when training the diffusion model. Then $L_{\text{VLB}}$ can be simplified as our score intrinsic reward $\mathcal{R}_{\text{score}}$. Thus, maximizing the score intrinsic reward can maximize the state entropy during the training of the diffusion model.

### C.2  ANALYSES OF THE OBJECTIVE $J_{\text{f}}$ WHEN FINE-TUNING DIFFUSION POLICIES

The final objective of the fine-tuning stage is to maximize $J$ of the policy $\pi$ within limited steps, and we may use any online RL methods like DDPG (Lillicrap, 2015), PPO (Schulman et al., 2017), and so on.

Unfortunately, when fine-tuning diffusion policies, directly using existing RL methods may encounter many challenges. For example, in the diffusion policies, the log p is hard to estimate (while log p is important in policy-gradient-based methods), and taking action from the diffusion policies requires the multi-step forward process of the neural network (so it is unstable to update the policy via $\nabla a\pi_\theta(s)$, like DDPG). There are several diffusion RL methods that hope to address these issues, like DQL (Wang et al., 2023), IDQL (Hansen-Estruch et al., 2023), QSM (Psenka et al., 2023), DIPO (Yang et al., 2023a), and so on. And we have included these methods as baselines in Fig. 3(c).

Given that the fine-tuning stage can only access the limited steps, we directly consider the optimal policy we can obtain, i.e., $\arg\max_\pi J_{\text{f}}(\pi)$, and hope to directly sample from the $\arg\max_\pi J_{\text{f}}(\pi)$ without using methods like policy gradient. (Thus, here the $\beta$ implicitly relies on the fine-tuning steps; if the fine-tuning step is infinity, the optimal we can access is $\arg\max_\pi J(\pi)$, i.e., $\beta = 0$; if the fine-tuning step is 0, the optimal we can access is $\pi_{\text{d}}$, i.e., $\beta = \infty$)

Given $J_{\text{f}}$, our results demonstrate that $\arg\max_\pi J_{\text{f}}(\pi)$ is the form of $\propto \pi_{\text{d}}e^{Q/\beta}$. This also provides another important insight: the more fine-tuning steps we can take, the closer we learn Q to the optimal Q function, and at this point, we should choose a smaller beta.

Thus, the core idea of ExDM is: learning the Q function, then utilizing guided sampling techniques to sample from $\propto \pi_{\text{d}}e^{Q/\beta}$, avoiding calculating $\log \pi(\boldsymbol{a}|\boldsymbol{s})$ or $\nabla_a\pi_\theta(s)$, which are difficult to estimate in diffusion policies. In practice, we find that distilling the pre-trained policy as well as the learned Q function into the fine-tuned policy can further improve the performance (this training is similar to the supervised training of the diffusion model, thus it is stable).

## C.3 Q FUNCTION OPTIMIZATION

For the Q function optimization, we choose to use implicit Q-learning (IQL) (Kostrikov et al., 2022), which is efficient to penalize out-of-distribution actions (Hansen-Estruch et al., 2023). The main training pipeline of IQL is expectile regression, i.e.,

$$
\begin{aligned}
\min_{\zeta} L_V(\zeta) &= \mathbb{E}_{\boldsymbol{s},\boldsymbol{a}\sim\mathcal{D}}\left[L_2^{\tau}(Q_{\phi}(\boldsymbol{s},\boldsymbol{a})-V_{\psi}(\boldsymbol{s}))\right], \\
\min_{\phi} L_Q(\phi) &= \mathbb{E}_{\boldsymbol{s},\boldsymbol{a},\boldsymbol{s}'\sim\mathcal{D}}\left[\|r(\boldsymbol{s},\boldsymbol{a})+\gamma V_{\zeta}(\boldsymbol{s}')-Q_{\phi}(\boldsymbol{s},\boldsymbol{a})\|^2\right],
\end{aligned}
\tag{32}
$$

here $L_2^{\tau}(\boldsymbol{u}) = |\tau - \mathbb{1}(\boldsymbol{u} < 0)|\boldsymbol{u}^2$ and $\tau$ is a hyper-parameter. In detail, when $\tau > 0.5$, $L_2^{\tau}$ will downweight actions with low Q-values and give more weight to actions with larger Q-values.

## C.4 DIFFUSION POLICY FINE-TUNING

For sampling from $\pi_n = \frac{1}{Z(s)}\pi_{\mathrm{d}}e^{Q_{n-1}/\beta}$, we choose contrastive energy prediction (CEP) (Lu et al., 2023), a powerful guided sampling method. First, we calculate the score function of $\pi_n$ as

$$
\nabla_{\boldsymbol{a}}\log\pi_n(\boldsymbol{a}|\boldsymbol{s}) = \nabla_{\boldsymbol{a}}\log\pi_{\mathrm{d}}(\boldsymbol{a}|\boldsymbol{s}) + \frac{1}{\beta}\nabla_{\boldsymbol{a}}Q_{n-1}(\boldsymbol{s},\boldsymbol{a}).
\tag{33}
$$

Moreover, to calculate the score function of $\pi_n$ at each timestep $t$, i.e., $\nabla_{\boldsymbol{a}_t}\log\pi_t^n(\boldsymbol{a}|\boldsymbol{s})$, CEP further defines the following Intermediate Energy Guidance:

$$
\mathcal{E}_t^{n-1}(\boldsymbol{s},\boldsymbol{a}_t) = \begin{cases} \frac{1}{\beta}Q_{n-1}(\boldsymbol{s},\boldsymbol{a}_0), & t = 0 \\ \log\mathbb{E}_{\mu_{0t}(\boldsymbol{a}_0|\boldsymbol{s},\boldsymbol{a}_t)}\left[e^{Q_{n-1}(\boldsymbol{s},\boldsymbol{a}_0)/\beta}\right], & t > 0 \end{cases}
\tag{34}
$$

Then Theorem 3.1 in CEP proves that

$$
\begin{aligned}
\pi_t^n(\boldsymbol{a}_t|\boldsymbol{s}) &\propto \pi_{\mathrm{d}}(\boldsymbol{a}_t|\boldsymbol{s})e^{\mathcal{E}_t^{n-1}(\boldsymbol{s},\boldsymbol{a}_t)}, \\
\nabla_{\boldsymbol{a}_t}\log\pi_t^n(\boldsymbol{a}_t|\boldsymbol{s}) &= \nabla_{\boldsymbol{a}_t}\log\pi_{\mathrm{d}}(\boldsymbol{a}_t|\boldsymbol{s}) + \nabla_{\boldsymbol{a}}\mathcal{E}_t^{n-1}(\boldsymbol{s},\boldsymbol{a}_t).
\end{aligned}
\tag{35}
$$

For estimating $\nabla_{\boldsymbol{a}}\mathcal{E}_t^{n-1}(\boldsymbol{s},\boldsymbol{a}_t)$, CEP considers a parameterized neural network $f_{\phi_{n-1}}(\boldsymbol{s},\boldsymbol{a}_t,t)$ with the following objective:

$$
\min_{\phi_{n-1}}\mathbb{E}_{t,\boldsymbol{s}}\mathbb{E}_{\boldsymbol{a}^1,\ldots,\boldsymbol{a}^K\sim\pi_{\mathrm{d}}(\cdot|\boldsymbol{s})}\left[-\sum_{i=1}^{K}\frac{e^{Q_{n-1}(\boldsymbol{s},\boldsymbol{a}^i)/\beta}}{\sum_{j=1}^{K}e^{Q_{n-1}(\boldsymbol{s},\boldsymbol{a}^j)/\beta}}\log\frac{f_{\phi_{n-1}}(\boldsymbol{s},\boldsymbol{a}_t^i,t)}{\sum_{j=1}^{K}f_{\phi_{n-1}}(\boldsymbol{s},\boldsymbol{a}_t^j,t)}\right].
\tag{36}
$$

Then Theorem 3.2 in CEP (Lu et al., 2023) has proven that its optimal solution $f_{\phi_{n-1}^*}$ satisfying that $\nabla_{\boldsymbol{a}_t}f_{\phi_{n-1}^*}(\boldsymbol{s},\boldsymbol{a}_t,t) = \nabla_{\boldsymbol{a}_t}\mathcal{E}_t^{n-1}(\boldsymbol{s},\boldsymbol{a}_t)$.

Consequently, we propose to fine-tune $\nabla_{\boldsymbol{a}_t}\log\pi_t^n(\boldsymbol{a}_t|\boldsymbol{s})$ parameterized as $\boldsymbol{s}_{\psi}(\boldsymbol{a}_t|\boldsymbol{s},t)$ with the following distillation objective:

$$
\min_{\psi}\mathbb{E}_{\boldsymbol{s},\boldsymbol{a},t}\|\boldsymbol{\epsilon}_{\psi}(\boldsymbol{a}_t|\boldsymbol{s},t) - \boldsymbol{\epsilon}_{\theta}(\boldsymbol{a}_t|\boldsymbol{s},t) - f_{\phi_{n-1}}(\boldsymbol{s},\boldsymbol{a}_t,t)\|^2.
\tag{37}
$$

And the optimal solution $\psi^*$ satisfying that $\boldsymbol{\epsilon}_{\psi^*}(\boldsymbol{a}_t|\boldsymbol{s},t)$ is the score function of $\pi_n$, i.e., we can sample from $\boldsymbol{\epsilon}_{\psi^*}(\boldsymbol{a}_t|\boldsymbol{s},t)$ with any unconditional diffusion model sampling methods like DDIM (Song et al., 2021a) or DPM-solver (Lu et al., 2022).

## D    EXPERIMENTAL DETAILS

In this section, we will introduce more information about our experimental details. In Sec. D.1, we first introduce all the domains and tasks evaluated in our experiments. Then we briefly illustrate all the baselines compared in experiments in Sec. D.2. Then in Sec. D.3, we introduce the hyperparameters of ExDM. Moreover, we supplement more detailed experimental results about maze2d and URLB in Sec. D.4 and Sec. D.5, respectively. The detailed ablation studies are in Sec. D.6. And we finally report the computing resource in Sec. D.7.

### D.1    DOMAINS AND TASKS

**Maze2d.**    This setting includes 7 kinds of mazes: Square-a, Square-b, Square-c, Square-d, Square-tree, Square-bottleneck, and Square-large. These mazes are two-dimensional, and agents need to explore as many states as possible during the unsupervised pre-training stage.

**Continuous Control.**    Our domains of continuous control follow URLB (Laskin et al., 2021), including 4 domains: Walker, Quadruped, Jaco, and Hopper, each with 4 downstream tasks from Deepmind Control Suite (DMC) (Tassa et al., 2018) (we plot each task of each domain in Fig. 5):

- **Walker** is a two-legged robot, including 4 downstream tasks: **stand**, **walk**, **run**, and **flip**. The maximum episodic length and reward for each task is 1000.
- **Quadruped** is a quadruped robot within a 3D space, including 4 tasks: **stand**, **walk**, **run**, and **jump**. The maximum episodic length and reward for each task is 1000.
- **Jaco** is a 6-DOF robotic arm with a 3-finger gripper, including 4 tasks: **reach-top-left (tl)**, **reach-top-right (tr)**, **reach-bottom-left (bl)**, and **reach-bottom-right (br)**. The maximum episodic length and reward for each task is 250.
- **Hopper** is a one-legged hopper robot, including 4 tasks: **hop**, **hop-backward**, **flip**, and **flip-backward**. The maximum episodic length and reward for each task is 1000.

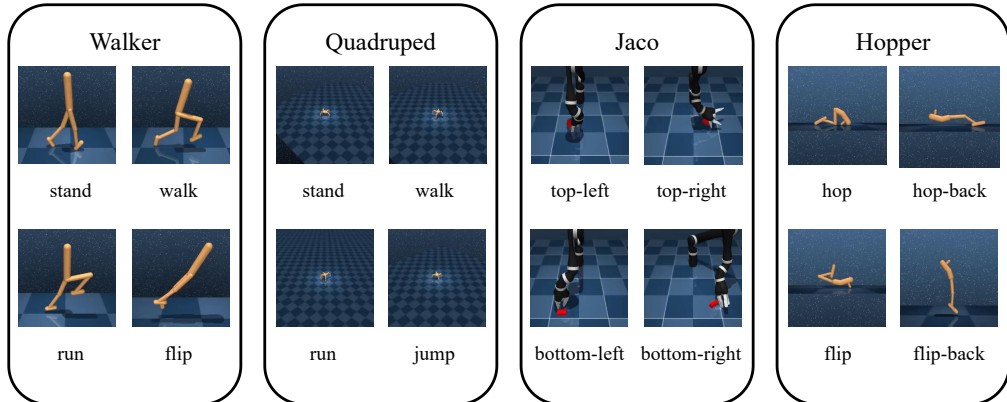

Figure 5: Illustration of domains with their downstream tasks in URLB (Laskin et al., 2021). We consider 4 domains, and each domain has four downstream tasks.

### D.2    BASELINES AND IMPLEMENTATIONS

We first introduce all URL baselines in our experiments.

**ICM (Pathak et al., 2017).**    Intrinsic Curiosity Module (ICM) trains a forward dynamics model and designs intrinsic rewards as the prediction error of the trained dynamics model.

**RND (Burda et al., 2018).**    Random Network Distillation (RND) utilizes the error between the predicted features of a trained neural network and a fixed randomly initialized neural network as the intrinsic rewards.

**Disagreement (Pathak et al., 2019)**   The Disagreement algorithm proposes a self-supervised algorithm that trains an ensemble of dynamics models and leverages the prediction variance between multiple models to estimate state uncertainty.

**LBS (Mazzaglia et al., 2022).**   Latent Bayesian Surprise (LBS) designs the intrinsic reward as the Bayesian surprise within a latent space, i.e., the difference between prior and posterior beliefs of system dynamics.

**DIAYN (Eysenbach et al., 2018).**   Diversity is All You Need (DIAYN) proposes to learn a diverse set of skills during the unsupervised pre-training stage, by maximizing the mutual information between states and skills.

**SMM (Lee et al., 2019).**   State Marginal Matching (SMM) aims at learning a policy, of which the state distribution matches a given target state distribution.

**LSD (Park et al., 2022).**   Lipschitz-constrained Skill Discovery (LSD) adopts a Lipschitz-constrained state representation function for maximizing the traveled distances of states and skills.

**CIC (Laskin et al., 2022).**   Contrastive Intrinsic Control (CIC) leverages contrastive learning between state and skill representations, which can both learn the state representation and encourage behavioral diversity.

**BeCL (Yang et al., 2023b).**   Behavior Contrastive Learning (BeCL) defines intrinsic rewards as the mutual information (MI) between states sampled from the same skill, utilizing contrastive learning among behaviors.

**CeSD (Bai et al., 2024).**   Constrained Ensemble exploration for Skill Discovery (CeSD) utilizes an ensemble of value functions for distinguishing different skills and encourages the agent to explore the state space with a partition based on the designed prototype.

**PEAC (Ying et al., 2024).**   Pre-trained Embodiment-Aware Control (PEAC) analyzes the fine-tuning and pre-training objectives for the cross-embodiment unsupervised RL, resulting the cross-embodiment intrinsic rewards.

In experiments of URLB, most baselines (ICM, RND, Disagreement, DIAYN, SMM) combined with RL backbone DDPG are directly following the official implementation in urlb (https://github.com/rll-research/url_benchmark). For LBS, we refer to the official implementation (https://github.com/mazpie/mastering-urlb) and combine it with the codebase of urlb. For CIC, BeCL, CeSD, and PEAC, we also follow their official implementations (https://github.com/rll-research/cic, https://github.com/Rooshy-yang/BeCL, https://github.com/Baichenjia/CeSD, https://github.com/thu-ml/CEURL), respectively.

Below, we will list the diffusion fine-tuning baselines in our experiments.

**DQL (Wang et al., 2023).**   Diffusion Q-Learning (DQL) inherited the idea of policy gradient and proposes to directly backpropagate the gradient of the Q function within the actions (calculated with the diffusion action by multi-step denoising).

**QSM (Psenka et al., 2023).**   Q-Score Matching (QSM) proposes to align the score of the diffusion policy with the gradient of the learned Q function.

**DIPO (Yang et al., 2023a).**   Diffusion Policy for Model-free Online RL (DIPO) utilizes the Q function to optimize the actions, i.e., finding the better action with gradient ascent of the Q function, and then trains the diffusion policy to fit the "optimized" actions.

**IDQL (Hansen-Estruch et al., 2023).** Implicit Diffusion Q-learning (IDQL) considers to sample multiple actions from the diffusion policy and then select the optimal action with the learned Q function.

We implement these methods based on their official codebases: DQL (`https://github.com/Zhendong-Wang/Diffusion-Policies-for-Offline-RL`), QSM (`https://github.com/Alescontrela/score_matching_rl`), DIPO (`https://github.com/BellmanTimeHut/DIPO`), and IDQL (`https://github.com/philippe-eecs/IDQL`).

### D.3 HYPERPARAMETERS

Hyperparameters of baselines are taken from their implementations (see Appendix D.2 above). Here we introduce ExDM's hyperparameters.

First, for the RL backbone DDPG, our code is based on URLB (`https://github.com/rll-research/url_benchmark`) and inherits DDPG's hyperparameters. For the diffusion model hyperparameters, we follow CEP (Lu et al., 2023). For completeness, we list all hyperparameters in Table 2.

| **DDPG Hyperparameter** | Value |
|---|---|
| Replay buffer capacity | $10^6$ |
| Action repeat | 1 |
| Seed frames | 4000 |
| n-step returns | 3 |
| Mini-batch size | 1024 |
| Seed frames | 4000 |
| Discount $\gamma$ | 0.99 |
| Optimizer | Adam |
| Learning rate | 1e-4 |
| Agent update frequency | 2 |
| Critic target EMA rate $\tau_Q$ | 0.01 |
| Features dim. | 1024 |
| Hidden dim. | 1024 |
| Exploration stddev clip | 0.3 |
| Exploration stddev value | 0.2 |
| Number of pre-training frames | $1\times10^5$ for Maze2d and $2\times10^6$ for URLB |
| Number of fine-turning frames | $1\times10^5$ for URLB |
| **ExDM Hyperparameter** | Value |
| Diffusion SDE | VP SDE |
| $\alpha_t$ of diffusion model | $\alpha_t = -\frac{\beta_1-\beta_0}{4}t^2 - \frac{\beta_0}{2}t, \beta_0 = 0.1, \beta_1 = 20$ |
| $\sigma_t$ of diffusion model | $\sigma_t = \sqrt{1-\alpha_t^2}$ |
| Diffusion model neural network | 3 MLPResnet Blocks, hidden_dim=256, the same as IDQL (Hansen-Estruch et al., 2023) |
| Optimizer | Adam |
| Learning rate | 1e-4 |
| Energy guidance model | 4-layer MLP, hidden_dim=256, the same as CEP (Lu et al., 2023) |
| Sampling method | DPM-Solver |
| Sampling step | 15 |

Table 2: Details of hyperparameters used for Maze2d and state-based URLB.

## D.4 Additional Experiments in Maze

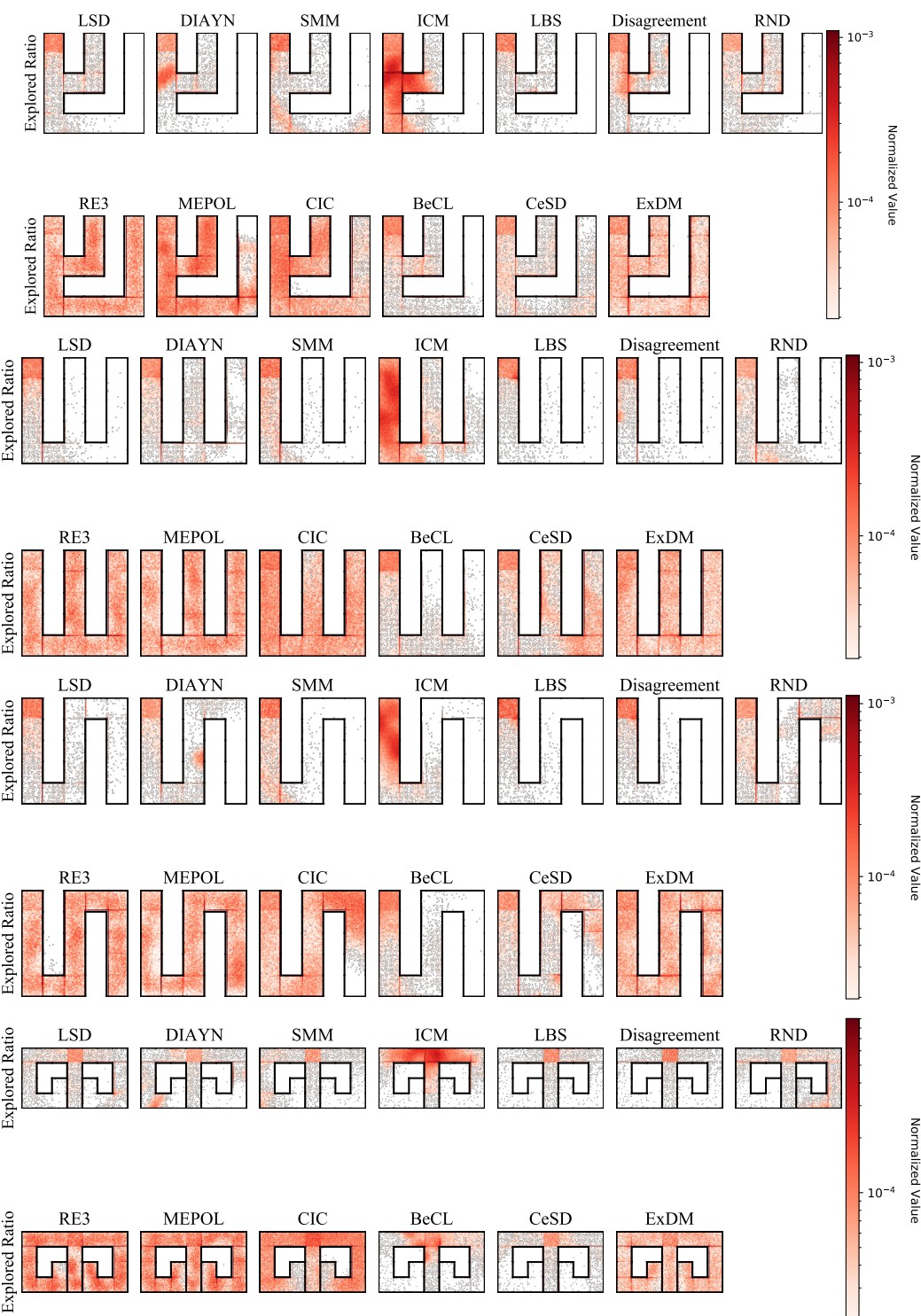

Figure 6: Visualization of explored trajectories by URL methods in **Square-a**, **Square-b**, **Square-c**, and **Square-d** maze.

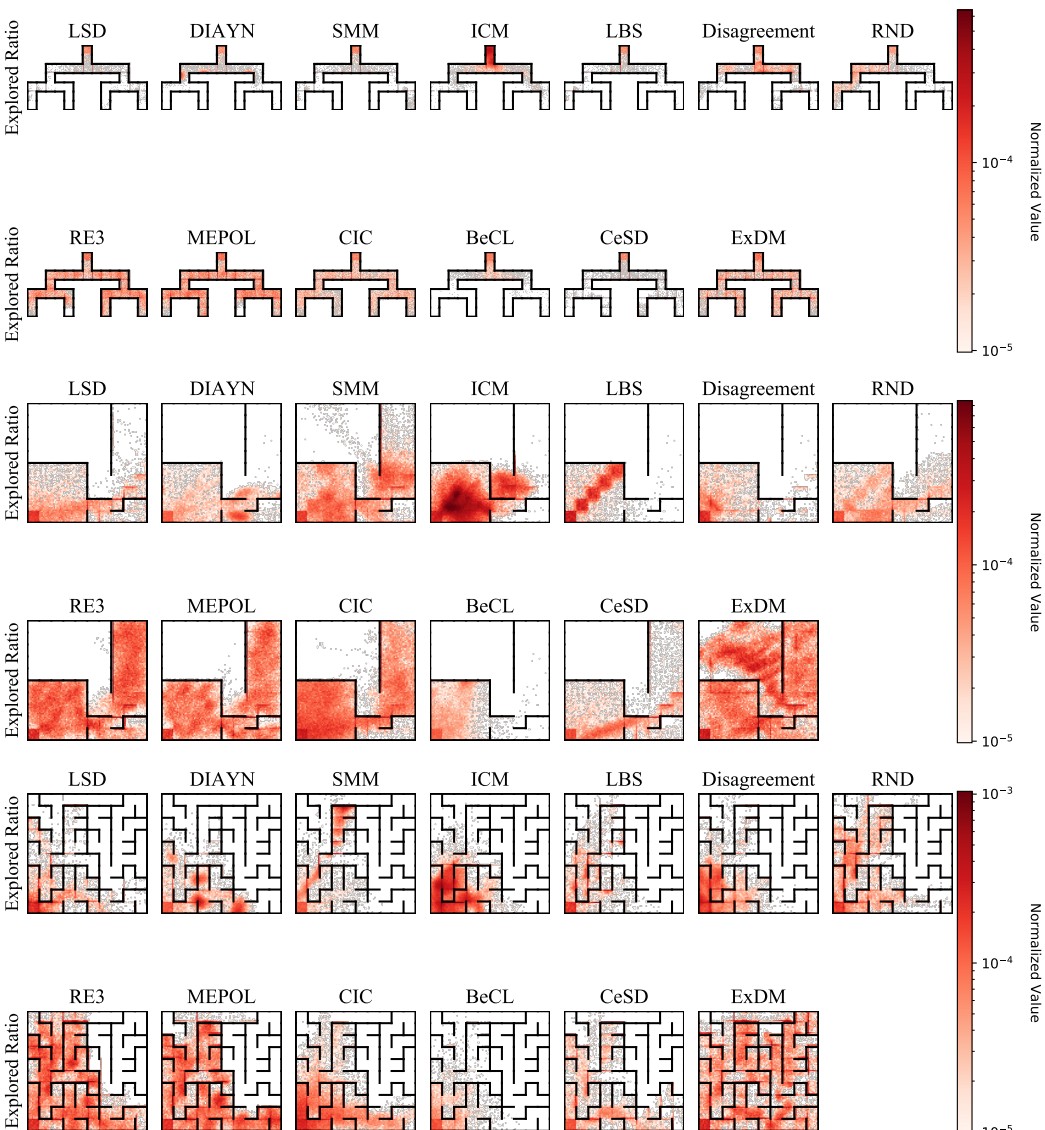

Figure 7: Visualizations of explored trajectories by URL methods in mazes **Square-tree**, **Square-bottleneck**, and **Square-large**, respectively.

Moreover, we include the visualization of all algorithms (ICM, RND, Disagreement, RE3, MEPOL, LBS, DIAYN, SMM, LSD, CIC, BeCL, CeSD, and ExDM) within all 7 mazes: Square-a, Square-b, Square-c, Square-d, Square-tree, Square-bottleneck, and Square-large, in Fig. 6 - Fig. 7, respectively.

As shown in these figures, although baselines can explore unseen states and try to cover as many states as they can, the behaviors of baselines can not fully cover the explored replay buffer due to their limited expressive ability. Using the strong modeling ability of diffusion models, the pre-trained policies of ExDM can perform diverse behaviors, setting a great initialization for handling downstream tasks.

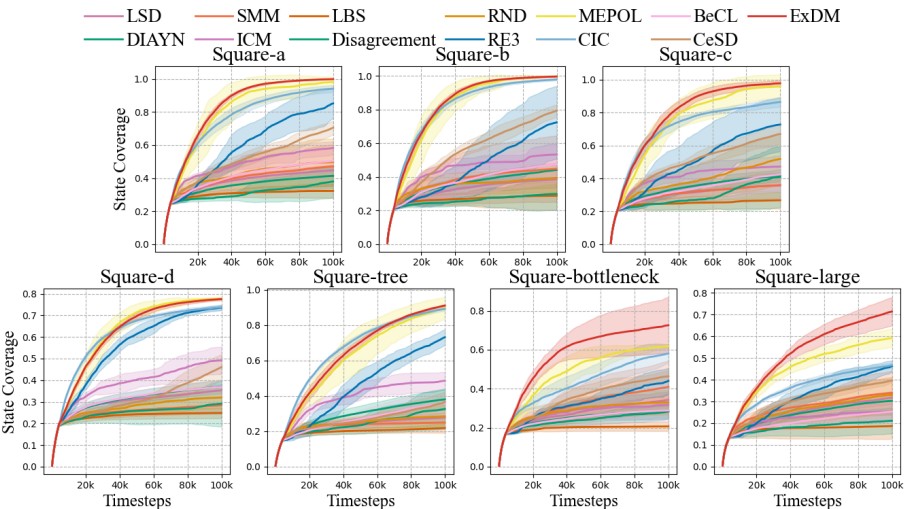

Figure 8: **State coverage ratios** of different algorithms in 7 mazes during pre-training.

## D.5 ADDITIONAL EXPERIMENTS IN URLB

In Table 3, we report the detailed results of all methods in the 4 downstream tasks of 4 domains in URLB. In both the Quadruped and Jaco domains, ExDM obtains state-of-the-art performance in downstream tasks. Overall, there are the most number of downstream tasks that ExDM performs the best, and ExDM significantly outperforms existing exploration algorithms.

| Domains | Walker | | | | Quadruped | | | | Jaco | | | | Hopper | | | |
|---|---|---|---|---|---|---|---|---|---|---|---|---|---|---|---|---|
| Tasks | stand | walk | run | flip | stand | walk | run | jump | tl | tr | bl | br | hop | hop-back | flip | flip-back |
| ICM | 828.5 | 628.8 | 223.8 | 400.3 | 298.9 | 129.9 | 92.1 | 148.8 | 96.5 | 91.7 | 84.3 | 83.4 | 82.1 | 160.5 | 106.9 | 107.6 |
| RND | 878.3 | 745.4 | 348.0 | 454.1 | 792.0 | 544.5 | 447.2 | 612.0 | 98.7 | 110.3 | 107.0 | 105.2 | 83.3 | **267.2** | 132.5 | 184.0 |
| Disagreement | 749.5 | 521.9 | 210.5 | 340.1 | 560.8 | 382.3 | 361.9 | 427.9 | 142.5 | 135.1 | 129.6 | 118.1 | 86.2 | 255.6 | 113.0 | 215.3 |
| LBS | 594.9 | 603.2 | 138.8 | 375.3 | 413.0 | 253.2 | 203.8 | 366.6 | 166.5 | 153.8 | 129.6 | 139.6 | 24.8 | 240.2 | 88.9 | 105.6 |
| APT | **953.3** | **900.0** | 504.1 | 675.6 | 582.0 | 582.0 | 303.0 | 416.0 | 76.7 | 116.2 | 121.1 | 121.1 | 133.3 | 260.2 | 135.7 | 202.7 |
| RE3 | 905.5 | 777.5 | 322.1 | 441.7 | 841.0 | 705.8 | 453.2 | 604.0 | 109.1 | 114.0 | 100.7 | 97.4 | 86.5 | 213.0 | 113.0 | 161.1 |
| MEPOL | 936.0 | 775.8 | 306.3 | 471.2 | 609.6 | 491.3 | 298.5 | 595.8 | 124.9 | 138.8 | 113.4 | 130.6 | 70.6 | 164.0 | 92.6 | 119.1 |
| DIAYN | 721.7 | 488.3 | 186.9 | 317.0 | 640.8 | 525.1 | 275.1 | 567.8 | 29.7 | 15.6 | 30.4 | 38.6 | 1.7 | 10.8 | 0.7 | 0.5 |
| SMM | 914.3 | 709.6 | 347.4 | 442.7 | 223.9 | 93.8 | 91.6 | 96.2 | 57.8 | 30.1 | 34.8 | 45.0 | 29.3 | 61.4 | 47.0 | 29.7 |
| APS | 575.8 | 472.5 | 155.4 | 374.4 | 484.8 | 335.6 | 387.8 | 351.7 | 34.0 | 40.0 | 29.2 | 43.8 | 1.0 | 2.0 | 3.0 | 10.0 |
| LSD | 770.2 | 532.3 | 167.1 | 309.7 | 319.4 | 186.3 | 179.6 | 283.5 | 11.6 | 33.6 | 22.5 | 6.7 | 12.0 | 6.6 | 2.9 | 12.2 |
| CIC | 941.1 | 883.1 | 399.0 | 687.2 | 789.1 | 587.8 | 475.1 | 630.6 | 148.8 | 167.6 | 122.3 | 145.9 | 82.7 | 191.6 | 96.2 | 161.3 |
| BeCL | 951.7 | 912.7 | 408.6 | 626.2 | 798.7 | 694.9 | 391.7 | 645.5 | 114.2 | 132.2 | 117.7 | 144.7 | 37.1 | 68.3 | 73.6 | 142.7 |
| CeSD | 884.0 | 838.7 | 325.2 | 570.9 | **886.5** | 763.4 | **636.4** | 759.1 | 155.7 | 170.2 | 137.3 | 117.9 | 118.4 | 155.7 | 46.4 | 183.6 |
| ExDM (Ours) | 905.9 | 874.1 | 389.5 | 572.1 | 915.4 | 873.6 | 569.5 | 755.2 | 173.5 | 202.6 | 170.3 | 170.5 | 107.6 | 275.8 | 155.3 | 220.1 |

Table 3: **Detailed results in URLB of different pre-trained methods that fine-tune Gaussian policies with DDPG**. Average cumulative reward (mean of 10 seeds) of the best policy.

| Metrics | Median | IQM | Mean | Optimality Gap |
|---|---|---|---|---|
| ICM | 0.40, [0.34, 0.44] | 0.38, [0.33, 0.42] | 0.39, [0.35, 0.43] | 0.61, [0.57, 0.65] |
| RND | 0.59, [0.55, 0.63] | 0.59, [0.56, 0.62] | 0.59, [0.56, 0.62] | 0.41, [0.38, 0.44] |
| Disagreement | 0.54, [0.50, 0.58] | 0.53, [0.49, 0.57] | 0.57, [0.51, 0.57] | 0.46, [0.43, 0.50] |
| LBS | 0.48, [0.41, 0.53] | 0.45, [0.39, 0.51] | 0.47, [0.43, 0.52] | 0.53, [0.49, 0.58] |
| RE3 | 0.46, [0.41, 0.51] | 0.46, [0.41, 0.51] | 0.46, [0.42, 0.51] | 0.54, [0.50, 0.58] |
| MEPOL | 0.48, [0.43, 0.53] | 0.48, [0.42, 0.53] | 0.48, [0.44, 0.53] | 0.52, [0.48, 0.56] |
| DIAYN | 0.29, [0.24, 0.34] | 0.24, [0.18, 0.30] | 0.29, [0.25, 0.33] | 0.71, [0.67, 0.75] |
| SMM | 0.29, [0.23, 0.34] | 0.21, [0.16, 0.27] | 0.29, [0.24, 0.33] | 0.71, [0.67, 0.76] |
| LSD | 0.21, [0.17, 0.26] | 0.14, [0.10, 0.19] | 0.21, [0.18, 0.25] | 0.79, [0.75, 0.82] |
| CIC | 0.66, [0.61, 0.70] | 0.68, [0.63, 0.72] | 0.66, [0.62, 0.69] | 0.35, [0.31, 0.38] |
| BeCL | 0.59, [0.54, 0.66] | 0.62, [0.56, 0.68] | 0.60, [0.55, 0.64] | 0.40, [0.36, 0.45] |
| CeSD | 0.67, [0.62, 0.72] | 0.71, [0.67, 0.76] | 0.67, [0.63, 0.71] | 0.33, [0.29, 0.37] |
| ExDM (Ours) | **0.78**, [0.74, 0.81] | **0.80**, [0.76, 0.84] | **0.77**, [0.74, 0.81] | **0.23**, [0.20, 0.26] |

Table 4: **Aggregate metrics (Agarwal et al., 2021) with confidence interval in URLB**. For every algorithm, there are 4 domains, each trained with 10 seeds and fine-tuned under 4 downstream tasks, thus each statistic for every method has 160 runs.

Moreover, we further provide the detailed metrics (including the confidence interval) of all methods within URLB in Table 4. As shown here, ExDM significantly outperforms all baselines, for example, ExDM's IQM is larger than the second-best method, CeSD, by 13%.

In Table 5, we further report the detailed results of all methods in the 4 downstream tasks of 2 domains in cross-embodiment URLB, which is much more challenging as the algorithms require handling various embodiments. In both the Walker-mass and Quadruped-mass domains, ExDM obtains state-of-the-art performance in downstream tasks. Overall, there are the most number of downstream tasks that ExDM performs the best, and ExDM significantly outperforms existing exploration algorithms.

| Domains | Walker-mass | | | | Quadruped-mass | | | |
| Tasks | stand | walk | run | flip | stand | walk | run | jump |
| --- | --- | --- | --- | --- | --- | --- | --- | --- |
| ICM | 665.3 | 418.0 | 146.2 | 246.6 | 460.2 | 229.5 | 215.6 | 323.5 |
| RND | 588.9 | 386.7 | 176.4 | 253.8 | 820.6 | 563.7 | 409.6 | 589.5 |
| Disagreement | 549.3 | 331.6 | 139.8 | 250.0 | 555.5 | 372.4 | 329.8 | 506.1 |
| PEAC | 823.8 | 499.9 | 210.6 | 320.5 | 786.0 | **754.5** | 388.3 | **645.6** |
| DIAYN | 502.1 | 245.2 | 106.8 | 212.7 | 682.7 | 484.3 | 371.0 | 469.1 |
| SMM | 673.5 | 509.2 | 220.7 | 329.6 | 357.0 | 176.4 | 189.7 | 277.8 |
| CIC | 824.8 | 536.6 | 220.7 | 327.7 | 762.5 | 610.9 | **442.7** | 617.5 |
| BeCL | **838.6** | **623.6** | 238.5 | 348.1 | 729.8 | 445.0 | 349.4 | 557.1 |
| ExDM (Ours) | **872.2** | **647.6** | **266.2** | **396.7** | **872.7** | **717.6** | **465.8** | **693.7** |

Table 5: **Detailed results in cross-embodiment state-based DMC**. Average cumulative reward (mean of 10 seeds) of the best policy.

| Metrics | Median | IQM | Mean | Optimality Gap |
| --- | --- | --- | --- | --- |
| ICM | 0.42, [0.37, 0.48] | 0.41, [0.35, 0.46] | 0.42, [0.38, 0.47] | 0.58, [0.53, 0.62] |
| RND | 0.57, [0.53, 0.62] | 0.57,[0.53, 0.61] | 0.58, [0.54, 0.62] | 0.42, [0.38, 0.46] |
| Disagreement | 0.47, [0.43, 0.50] | 0.45, [0.42, 0.48] | 0.47, [0.43, 0.50] | 0.53, [0.50, 0.57] |
| LBS | 0.56, [0.50, 0.60] | 0.52, [0.48, 0.57] | 0.55, [0.50, 0.59] | 0.45, [0.41, 0.50] |
| PEAC | 0.68, [0.63, 0.74] | 0.70, [0.64, 0.76] | 0.68, [0.63, 0.72] | 0.32, [0.28, 0.37] |
| DIAYN | 0.46, [0.40, 0.51] | 0.42,[0.38, 0.47] | 0.46, [0.42, 0.50] | 0.54, [0.50, 0.58] |
| SMM | 0.44, [0.39, 0.52] | 0.45, [0.37, 0.53] | 0.45, [0.40, 0.51] | 0.55, [0.49, 0.60] |
| CIC | 0.67, [0.62, 0.73] | 0.68, [0.62, 0.74] | 0.67, [0.63, 0.71] | 0.33, [0.29, 0.37] |
| BeCL | 0.65, [0.58, 0.71] | 0.67, [0.60, 0.73] | 0.65, [0.60, 0.70] | 0.35, [0.30, 0.40] |
| ExDM (Ours) | **0.77**, [0.73, 0.81] | **0.80**, [0.75, 0.83] | **0.77**, [0.74, 0.80] | **0.23**, [0.20, 0.26] |

Table 6: **Aggregate metrics (Agarwal et al., 2021) with confidence interval in cross-embodiment URLB**. For every algorithm, there are 2 domains, each trained with 10 seeds and fine-tuned under 4 downstream tasks; thus, each statistic for every method has 80 runs.

Moreover, we further provide the detailed metrics (including the confidence interval) of all methods within URLB in Table 6. As shown here, ExDM significantly outperforms all baselines, for example, ExDM's IQM is larger than the second-best method, PEAC, by 14%.

### D.6 ABLATION OF TIMESTEPS IN URLB

In Figure 9, we show additional results about the performance in four domains of URLB for different algorithms and pre-training timesteps. Overall, ExDM outperforms all methods, while CIC and CeSD are still competitive on some domains.

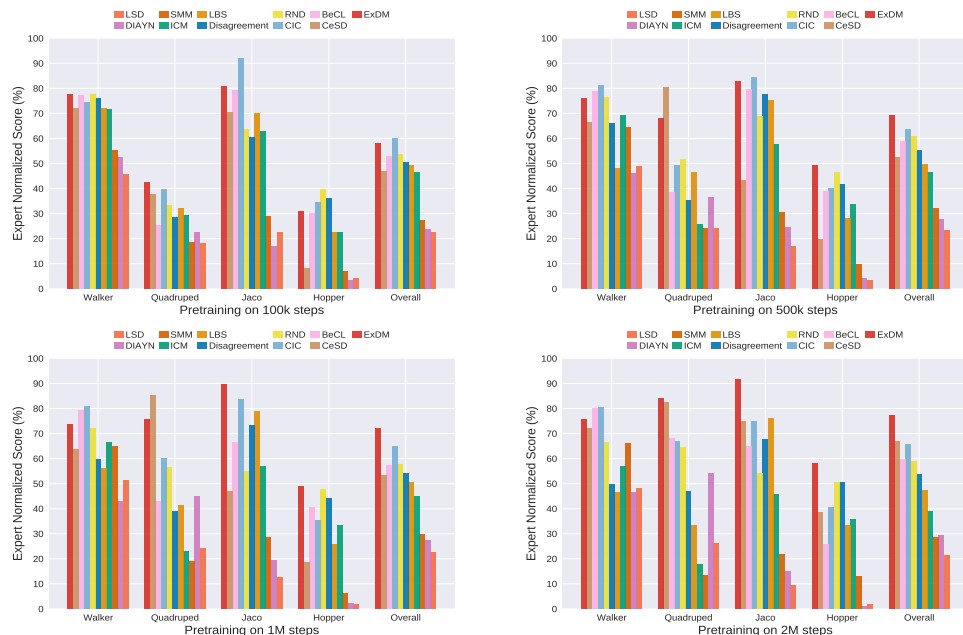

Figure 9: **Ablation study of pre-training steps in URLB**.

## D.7 COMPUTING RESOURCE

In experiments, all agents are trained by GeForce RTX 2080 Ti with Intel(R) Xeon(R) Silver 4210 CPU @ 2.20GHz. In maze2d / urlb, pre-training ExDM (each seed, domain) takes around 0.5 / 2 days, respectively.

## E THE USE OF LARGE LANGUAGE MODELS

In the preparation of this manuscript, LLMs were used solely as auxiliary tools for paraphrasing and polishing the writing to improve readability. No LLM was involved in formulating research ideas, designing methods, constructing datasets, implementing experiments, conducting analyses, or drawing conclusions. All scientific contributions and substantive content presented in this paper are the original work of the authors.

