# OpenReview forum: "Exploratory Diffusion Model for Unsupervised Reinforcement Learning"
_ICLR.cc/2026/Conference — ICLR 2026 Oral_

### Official Review · Reviewer_UWWM · 2025-10-25

**Soundness:** 2
**Presentation:** 3
**Contribution:** 2
**Rating:** 6
**Confidence:** 3

**Summary:**

This work proposes a novel approach called ExDM for unsupervised RL using a diffusion model. During pre-training, ExDM trains a diffusion model to model the state and action distributions of interactions with the environment and derives an intrinsic reward to encourage exploration that is inversely proportional with approximate probability of state visitations of the diffusion model. Using these intrinsic rewards, the approach trains a Gaussian policy that explores the environment. During fine-tuning, the Gaussian policy can be trained with task-specific rewards, or the diffusion policy trained during pre-training can be fine-tuned. To enable fine-tuning of the diffusion policy, a novel regularized training objective is being derived, similar to soft RL and using implicit Q-learning from the offline RL literature to avoid out-of-distribution actions being sampled. The new approach is shown to lead to more exhaustive exploration, as measured by state coverage, during pre-training, and the work shows that fine-tuning of the pre-trained Gaussian and diffusion policies lead to higher performance compared to alternative pre-training approaches.

**Strengths:**

## Originality
The approach proposed in this work appears original and novel. While diffusion policies are not new, and diffusion models have been used to express various data distributions, their application to URL is novel to the best of my knowledge. Furthermore, the theoretical contributions in Theorem 4.1 justifying the need for more expressive policies for unsupervised pre-training, and in deriving a novel algorithm for online fine-tuning of the diffusion policy are valuable to the community.

## Significance
As stated, I consider the theoretical contributions of this work significant and valuable to the community. Similarly, the empirical results indicate a small but consistent improvement of ExDM compared to the strongest URL baselines. Assuming these results were generated under fair hyperparameter tuning (see question 3), they demonstrate that ExDM is a significant contribution to the field.

One aspect that dampens my otherwise very positive impression of the significance of this work is the unclear benefits of the diffusion policy part of the ExDM algorithm. The diffusion policy is arguably the biggest and most complex novel contribution of this work, but it appears to not contribute meaningfully to the performance of the approach (see Weakness 1.). Without the diffusion policy, the ExDM algorithm could have also "just" been a diffusion model of the state distribution to derive a slightly novel intrinsic reward with a Gaussian policy. I would expect further discussion with reviewers and the authors to clarify that part of this work.

## Quality & Clarity
I find the writing and presentation of this work of a high quality. The empirical evaluation also appears to follow good practice, and provides further ablations and analyzes to shed more light on the learned components. There are few unclear or not well supported statements in this work that are listed below, but none of them are major issues or central to the work.

**Weaknesses:**

Below, I provide a list of weaknesses that should be addressed. I would consider all weaknesses stated as **major** important to address in order to justify acceptance.

1. **(Major)** The introduction of this work significantly leans into the motivation that typical URL approaches use policies that are not sufficiently expressive (often discrete or Gaussian policies) to properly explore the environment during pre-training. Theorem 4.1 as theoretical contribution of this work further supports this narrative and the pre-training and fine-tuning of the diffusion policy component of ExDM takes over large parts of Section 4. However, despite this motivation and more expressive diffusion policy, the environment interactions during pre-training are still done only using the Gaussian policy (as per line 8 in Algorithm 1). Furthermore, fine-tuning of the Gaussian policy of ExDM still leads to higher performance than fine-tuning the diffusion policy (see Figure 3 (a) vs (c)), a fact that is acknowledged by the authors in Section 5.4.
   All this makes me question what the diffusion policy of ExDM truly adds to the method. It appears the benefits of ExDM are not from training a more expressive policy in the diffusion policy, but from the diffusion model of state distributions that appears to provide a more informative intrinsic reward to exhaustively explore the environment. Theorem 4.1 still motivates the use of more expressive policies but it appears that the challenges of training diffusion policies remain to outweigh the benefits of a more expressive policy. My questions here are
	1. Does the diffusion policy of ExDM provide some benefits, or does it contribute to any component of the approach in a way that I have not seen?
	2. Or am I correct in my assessment that it appears to provide no benefits over the Gaussian policy so far?
	3. Given Theorem 4.1, one might expect the more expressive diffusion policy of ExDM to provide clear benefits. In Section 5.4, it is stated that the diffusion policy might perform worse than the Gaussian policy "[...] due to limited interaction timesteps during fine-tuning". Did you fine-tune the diffusion policies with the same 2M steps as used for Gaussian policy fine-tuning? If not, how many steps were used to fine-tune diffusion policies and why was a different number of fine-tuning steps used for these policies?
2. **Major:** Appendix C.3 states that "hyperparameters of baselines are taken from their implementations". Were these hyperparameters explicitly tuned for the Maze2D tasks evaluated in this work? How about ExDM hyperparameters listed in Table 2? Were these tuned in any form for the Maze2D tasks? I would expect comparable effort to be spent on tuning hyperparameters across all approaches to have confidence in the empirical results presented in this work, and this should be clarified.
3. The work makes the following two statements that are unclear, imprecise, or incorrect and should likely be expressed more precisely:
	1. "[...] the optimal policy of standard RL is a simple deterministic policy" (Section 4.1). It is unclear to me what "standard RL" means; to the best of my knowledge, the statement is only generally true in fully observable (single-agent) RL; partially observable or multi-agent environments might have stochastic optimal policies.
	2. "[...] the explored replay buffer is always diverse and heterogeneous, as the policy continuously changes to visit new states during pre-training. Consequently, URL requires capturing the heterogeneous distribution of collected data and obtaining policies with high diversity" -- I would argue that URL does not necessarily require to capture distribution of multiple policies to achieve its objective of learning a policy that maximizes state entropy (as stated in Section 4.1). It is merely the practical approach of using off-policy algorithms and a replay buffer of experiences that requires to capture such distributions.
	3. "The Gaussian behavior policy $\pi_g$ can then be trained using any RL algorithm" -- I would argue that this is not true since off-policy RL algorithms would be needed. The experience samples that $\pi_g$ is trained on, as per Algorithm 1, are from a replay buffer and, thus, off-policy with respect to the trained policy $\pi_g$.
4. The baselines visualized in Figure 2 appear to be mostly poor performing or middle of the pack when looking at Table 1. None of the strongest baselines (MEPOL, RE3, CIC) are included in Figure 2, supposedly to make the result of ExDM appear more impressive. I would appreciate Figure 2 would show the strongest 1-2 baselines in each family which appear to be R3 and MEPOL for exploration and CiC for skill discovery baselines. (Visualizations of all baselines are shown in Appendix C.4 but I would prefer for the most relevant ones to be shown in the main corpus of the paper)
5. The fine-tune box of Figure 1 appears confusing to me and I believe the policy titles should be flipped. The left half appears to show the fine-tuning of the Gaussian policy and the right half the fine-tuning of the diffusion policy (as per plot and legend) but the red titles above them are reversed.
6. (Minor) I noticed that baseline algorithms do not have identical colors in Figure 3 (a) and (b) which makes it slightly harder to cross-reference these results at a glance.

**Questions:**

1. What are the benefits of the diffusion policy of ExDM, or does it not contribute to the performance of the system (given the fine-tuned Gaussian policy performs better)?
2. Would the authors be able to elaborate on their statement that the diffusion policy might perform worse than the Gaussian policy "[...] due to limited interaction timesteps during fine-tuning (Section 5.4). Were the diffusion policies fine-tuned with less than the 2M steps used for Gaussian policies? What other reasons might there be for the more expressive diffusion policies to perform worse after fine-tuning?
3. How were hyperparameters of all baselines and ExDM selected for the empirical Maze2D experiments?

---

> ### Author Response · Authors · 2025-11-20
> **Thanks for your supportive comments (1/2)**
>
> Thank you so much for your detailed, kind, and supportive comments, which are really important to help us further polish our work and better position the contribution of this work. We'd like to address your concerns below:
>
> **W1, Q1-Q2: about the performance of diffusion policy fine-tuning in ExDM and the contribution of this work**
>
> **A:** We thank the reviewer for the kind suggestion to discuss the diffusion policy fine-tuning performance of ExDM. As unsupervised RL includes two important parts: **unsupervised exploration** and **downstream fine-tuning**, we think that fixing the algorithm of one part and modifying the algorithm of the other part for comparison can provide the fairest comparison. Consequently, we consider two settings in our experiments: (1) pre-training Gaussian policies with ExDM (compared with existing unsupervised RL methods like RND and CIC) and fine-tuning Gaussian policies with DDPG; (2) pre-training diffusion policies with ExDM and fine-tuning diffusion policies with ExDM (compared with other diffusion policy fine-tuning methods like IDQL and DIPO). We first hope to emphasize that
>
> - The results in setting (1) demonstrate that the unsupervised exploration part of ExDM has explored more diverse trajectories compared with existing baselines and obtains better downstream fine-tuning performance.
> - The results in setting (2) demonstrate that the diffusion policy fine-tuning of ExDM outperforms existing diffusion policy fine-tuning methods, such as IDQL and DIPO.
> - The diffusion policy fine-tuning performance of ExDM (Ours in setting (2)) is better than or comparable with the Gaussian policy fine-tuning performance of existing unsupervised RL methods (baselines in setting (1)) like RND or CIC.
> - The pre-trained diffusion policies can capture the diverse trajectories collected during pre-training and perform diversity.
>
> Moreover, as we have discussed in Sec 4.1, the diffusion policy fine-tuning performance of ExDM (Ours in setting (2)) is slightly worse than the Gaussian policy fine-tuning performance of ExDM (Ours in setting (1)).
> Our initial explanation in the manuscript pointed to the limited interaction steps during fine-tuning (here, limited means that the fine-tuning timesteps are always 1/10 or 1/20 of the pre-training steps).
> Below, we will explain it in detail.
>
> Although diffusion policies have been proven as expressive models for capturing diverse offline datasets, online fine-tuning diffusion policies still face many challenges and are a key issue of concern for the community, for example, the instability caused by the multi-step sampling and the lack of closed-form probability calculation.
> Consequently, the diffusion policy inherently requires more interaction data to fine-tune effectively and may converge more slowly in a low-data regime (limited interaction steps).
> Although many recent works hope to address these challenges and improve the efficiency of online training diffusion policies, they mainly consider two settings: **online training diffusion policies from scratch with enough timesteps** or **online fine-tuning diffusion policies pre-trained from an offline dataset sampled from the same task**.
> In this work, our setting is more challenging for fine-tuning diffusion policies, i.e., we need to **online fine-tune diffusion policies to downstream tasks that are unseen during pre-training with limited timesteps**.
> Our experiments demonstrate that existing online diffusion methods perform poorly in this challenging setting, and ExDM outperforms these baselines.
>
> Finally, we hope to emphasize our contributions, for both the pre-training stage and the fine-tuning stage, below:
>
> - During unsupervised exploration, ExDM introduces diffusion models to capture the diverse and heterogeneous trajectories collected in the replay buffer.
> - ExDM also pre-trains the diffusion policy to perform diversity. To fine-tune the pre-trained diffusion policy, we formulate the online fine-tuning objective and propose an alternating optimization, of which the optimality is guaranteed by Theorem 4.2. Its fine-tuning performance is better than existing diffusion policy fine-tuning methods (also better or comparable with existing unsupervised baselines with Gaussian policies).
>
> We hope our work and this challenging setting can inspire more research on online fine-tuning diffusion policies in unsupervised RL.
> Thanks again for your kind suggestions. This discussion will better position our work within the broader literature and outline clear future research directions.

---

> ### Author Response · Authors · 2025-11-20
> **Thanks for your supportive comments (2/2)**
>
> **W2, Q3: about hyperparameters of baselines and ExDM in Maze2d, are they tuned?**
>
> **A:** Thanks for your question. For all baselines and ExDM, we take the same hyperparameters for both Maze2d environments and URLB. In detail, for ICM, RND, Disagreement, DIAYN, and SMM, we use the default hyperparameters in the GitHub repository URLB (https://github.com/rll-research/url_benchmark).
> For LBS, RE3, MEPOL, and LSD, which have not been evaluated in URLB and Maze2d, we implement their algorithms based on the GitHub repository URLB and follow the hyperparameters of URLB settings and their papers.
> For CIC, BeCL, and CeSD, of which the code is based on the URLB, we directly utilize their hyperparameters from the GitHub repositories (https://github.com/rll-research/cic, https://github.com/Rooshy-yang/BeCL, https://github.com/Baichenjia/CeSD).
> As for the hyperparameters of ExDM, we have reported them in Table 2 of Appendix A.3. Specifically, for the basic hyperparameters of DDPG like discount, learning rate, optimizer, hidden dimension, critic target EMA rate, and so on, we keep them the same as URLB, which are the same for all algorithms.
> As for hyperparameters of diffusion models, we simply tune some of them, like sampling step, guidance scale, and so on, some of which have also undergone ablation experiments in Sec. 5.5.
>
> **W3: There are several statements that are unclear, imprecise, or incorrect and should likely be expressed more precisely:**
> - **"[...] the optimal policy of standard RL is a simple deterministic policy" (Section 4.1). It is unclear to me what "standard RL" means; to the best of my knowledge, the statement is only generally true in fully observable (single-agent) RL; partially observable or multi-agent environments might have stochastic optimal policies.**
> - **"[...] the explored replay buffer is always diverse and heterogeneous, as the policy continuously changes to visit new states during pre-training. Consequently, URL requires capturing the heterogeneous distribution of collected data and obtaining policies with high diversity" -- I would argue that URL does not necessarily require to capture distribution of multiple policies to achieve its objective of learning a policy that maximizes state entropy (as stated in Section 4.1). It is merely the practical approach of using off-policy algorithms and a replay buffer of experiences that requires to capture such distributions.**
> - **"The Gaussian behavior policy $\pi_{\mathrm{g}}$ can then be trained using any RL algorithm" -- I would argue that this is not true since off-policy RL algorithms would be needed. The experience samples that $\pi_{\mathrm{g}}$ is trained on, as per Algorithm 1, are from a replay buffer and, thus, off-policy with respect to the trained policy $\pi_{\mathrm{g}}$.**
>
> **A:** Thanks for your kind suggestions.
>
> (1): Yes, the ``standard RL" here represents fully observable (single-agent) RL, and we have clarified it in Sec. 4.1 of the revised version.
>
> (2): Yes, what we want to express here is that, even if the base policy is a Gaussian policy, it will keep updating during the pre-training stage, and the explored trajectories in the replay buffer will be diverse and heterogeneous, which need to be handled during the unsupervised pre-training stage. As you have stated, it is more like in practice, unsupervised RL needs to handle this diverse and heterogeneous distribution of the replay buffer. We have modified it in Sec. 4.1 of the revised version.
>
> (3): Yes, to better capture all historical trajectories the agent has explored, and improve the sample efficiency, off-policy RL is more suitable than on-policy RL in unsupervised RL. And we can use any off-policy RL algorithm during pre-training. We have modified it in Sec. 4.1 and Algorithm 1 of the revised version.
>
> **W4: The strongest baselines (MEPOL, RE3, CIC) should be included in Figure 2**
>
> **A:** Thanks for your suggestion. We hope to clarify that CIC is included in Fig.2. Besides CIC, we choose RND as it is one of the most classical exploration methods, and choose CeSD as it is one of the most recent baselines that chooses an ensemble of skill-based policies. As you have mentioned, we have reported the results of MEPOL and RE3 in our Appendix C.4. Following your constructive suggestions, we have supplemented their results in Fig.2 of the revised PDF to make it clearer.
>
> **W5: The fine-tuning part of Figure 1 should be modified.**
>
> **A:** Thanks a lot for your comment. There is a typo in Fig. 1; we have corrected it in the revised PDF.
>
> **W6: The baseline algorithms should have the same colors in Figure 3 (a) and (b).**
>
> **A:** Thanks for your kind suggestions. We agree that using the same color for the same algorithm in different figures will enhance the readability of the paper. Thus, we have modified it in Fig. 3 (b) of the revised version, to keep the same algorithm has the same color in Fig. 3 (a) and Fig. 3 (b).

---

> > ### Comment · Reviewer_UWWM · 2025-11-24
> > **Rebuttal Response**
> >
> > I thank the authors for their detailed rebuttal and revisions of their work. Their comments clarify all my major concerns and I decided to raise my score to 8: accept. In particular, I appreciate their more detailed discussion about the integration and relevance of their diffusion policy approach.
> >
> > I would suggest the authors to consider highlighting this separation of pre-training and fine-tuning the Gaussian and diffusion policies in the introduction, and to contextualize the results of their diffusion policy approach in more detail in Section 5.4.

---

> > > ### Author Response · Authors · 2025-11-25
> > > **Thanks again for your kind response**
> > >
> > > Dear reviewer UWWM,
> > >
> > > Thank you very much for your valuable suggestions and for increasing the score. We will try our best to further improve our paper in our next revised version to better clarify our contribution, especially about the separation of pre-training and fine-tuning the Gaussian and diffusion policies.
> > >
> > > Best regards,
> > >
> > > Authors

---

### Official Review · Reviewer_dZCp · 2025-10-27

**Soundness:** 3
**Presentation:** 3
**Contribution:** 3
**Rating:** 6
**Confidence:** 3

**Summary:**

The authors proposed an unsupervised RL algorithm called ExDM with diffusion action head.

**Strengths:**

1. The performance seems to be very strong compared to baselines
2. The presentation is clear and easy to follow

**Weaknesses:**

1. The motivation is somewhat weak. I'll put my questions in the following section.

**Questions:**

1. The results on URLB seems to be promising, but why APT and APS by Liu et al. were not included in the baseline?

---

> ### Author Response · Authors · 2025-11-20
> **Thanks for your supportive comments**
>
> Thanks a lot for your supportive comments. We will address your concerns below.
>
> **W1: The motivation is somewhat weak. I'll put my questions in the following section.**
>
> **A:** Thanks for your question. This work considers unsupervised RL, which is an important setting for improving the generalization of RL agents. In unsupervised RL, one of the major concerns is that the design of intrinsic rewards requires an accurate estimation of the explored trajectories in the replay buffer, which is always diverse and heterogeneous. Consequently, one of the most important motivations of this work is to first introduce diffusion models into unsupervised RL to capture the diversity during the unsupervised pre-training stage.
>
> **Q1: The results on URLB seems to be promising, but why APT and APS by Liu et al. were not included in the baseline?**
>
> **A:** Thanks for your question. APT and APS are two important entropy-based unsupervised RL methods, and we have discussed them in the related work. We have also compared other entropy-based baselines like RE3 and MEPOL in our experiments. Following your constructive suggestions, we have supplemented APT and APS as our baselines in fine-tuning the downstream tasks in URLB:
>
> ||walker-stand|walker-walk|walker-run|walker-flip|quadruped-stand|quadruped-walk|quadruped-run|quadruped-jump|jaco-tl|jaco-tr|jaco-bl|jaco-br|hopper-hop|hopper-hop-back|hopper-flip|hopper-flip-back|
> |-|-|-|-|-|-|-|-|-|-|-|-|-|-|-|-|-|
> |ICM|828.5|628.8|223.8|400.3|298.9|129.9|92.1|148.8|96.5|91.7|84.3|83.4|82.1|160.5|106.9|107.6|
> |RND|878.3|745.4|348.0|454.1|792.0|544.5|447.2|612.0|98.7|110.3|107.0|105.2|83.3| **267.2**| 132.5| 184.0|
> |Disagreement|749.5|521.9|210.5|340.1|560.8|382.3|361.9|427.9|142.5|135.1|129.6|118.1|86.2|255.6|113.0|**215.3**|
> |LBS|594.9|603.2|138.8|375.3|413.0|253.2|203.8|366.6|166.5|153.8|129.6|139.6|24.8|240.2|88.9|105.6|
> |RE3|905.5|777.5|322.1|441.7|841.0|705.8|453.2|604.0|109.1|114.0|100.7|97.4|86.5|213.0|113.0|161.1|
> |MOPEL|**936.0**|775.8|306.3|471.2|609.6|491.3|298.5|595.8|124.9|138.8|113.4|130.6|70.6|164.0|92.6|119.1|
> |DIAYN| 721.7|488.3 | 186.9 |317.0| 640.8| 525.1|275.1 |567.8|29.7|15.6 |30.4|38.6|1.7|10.8|0.7|0.5|
> |SMM| 914.3|709.6|347.4|442.7|223.9|93.8|91.6| 96.2| 57.8|30.1|34.8 |45.0|29.3|61.4 |47.0|29.7|
> |LSD|770.2|532.3|167.1 | 309.7|319.4 |186.3 |179.6 |283.5|11.6|33.6|22.5 |6.7|12.0|6.6| 2.9 |12.2|
> |CIC|**941.1**|**883.1**|**399.0**|**687.2**|789.1|587.8|475.1|630.6|148.8|167.6|122.3|145.9|82.7|191.6|96.2|161.3|
> |BeCL| **951.7** | **912.7** | **408.6** | 626.2| 798.7 | 694.9 |391.7 | 645.5| 114.2 | 132.2 | 117.7 | 144.7| 37.1 | 68.3 | 73.6 | 142.7|
> |CeSD| 884.0 | 838.7 | 325.2 | 570.9 | **886.5** | 763.4 | **636.4** | **759.1** | 155.7 | 170.2 | 137.3 | 117.9  | 118.4 | 155.7 | 46.4 | 183.6 |
> |APT |**953.3** | **900.0** | **504.1** | **675.6** | 582.0 | 582.0 | 303.0 | 416.0 | 76.7 | 116.2 | 121.1| 121.1 | **133.3**| **260.2** | 135.7| 202.7|
> |APS | 575.8 | 472.5 | 155.4 | 374.4 | 484.8 | 335.6 | 387.8 |351.7 | 34.0 | 40.0 | 29.2 | 43.8 | 1.0 | 2.0 | 3.0 | 10.0|
> |ExDM|905.9|**874.1**|**389.5**|572.1|**915.4**|**873.6**|**569.5**|**755.2**|**173.5**|**202.6**|**170.3**|**170.5**|107.6|**275.8**|**155.3**|**220.1**|
>
> As shown in these tables, state entropy-based methods like APT perform well in these benchmarks, as the estimated entropy provided good guidance for the agent to explore novel states. Still, ExDM outperforms these baselines due to the powerful fitting ability of diffusion models.

---

### Official Review · Reviewer_ueyP · 2025-10-31

**Soundness:** 3
**Presentation:** 3
**Contribution:** 3
**Rating:** 6
**Confidence:** 3

**Summary:**

This paper introduce the Exploratory Diffusion Model, which leverages the expressive power of diffusion models to fit diverse replay-buffer distributions, thus providing accurate density estimates and a score-based intrinsic reward that drives exploration into under-visited regions. This mechanism substantially broadens state coverage and yields robust pre-trained policies. Beyond exploration, ExDM develops an efficient decoupled training scheme and a finetuning algorithm for adapting pre-trained diffusion components to downstream tasks under limited interaction, with theoretical guarantees of convergence and optimality.

**Strengths:**

- Empirical gains across multiple settings: The figure indicates consistent improvements over strong unsupervised exploration baselines in URL, in cross-embodiment transfer, and when initializing diffusion policies.
- Potentially general mechanism: A diffusion-based exploratory prior could be a broadly applicable way to induce diverse skills or state coverage that helps downstream RL fine-tuning and transfer.
- Sufficient theoretical proof.

**Weaknesses:**

None

**Questions:**

- How is the intrinsic reward designed, what is the underlying rationale, and were alternative design schemes considered?
- What is the difference between unsupervised reinforcement learning and Meta-RL, given that both seem to aim at rapidly solving new tasks?

---

> ### Author Response · Authors · 2025-11-20
> **Thanks for your supportive comments**
>
> Thanks a lot for your supportive comments. We will address your concerns below.
>
> **Q1: How is the intrinsic reward designed, what is the underlying rationale, and were alternative design schemes considered?**
>
> **A:** Thanks for your question. In unsupervised RL, we pre-train agents in reward-free environments without downstream tasks. Thus, unsupervised RL algorithms require designing intrinsic rewards to encourage the agent to explore the environment as much as possible. One of the major rationales for intrinsic rewards is: assigning lower rewards to states that appear frequently in the replay buffer, and higher rewards to states that appear less frequently. Consequently, estimating the density of state distribution in the replay buffer is particularly important for designing intrinsic rewards. Our work first introduces diffusion models to design intrinsic rewards and outperforms existing unsupervised RL algorithms. Besides intrinsic rewards, there are several other unsupervised RL methods, like in-context RL or forward-backward representations, which have also been discussed in Sec. 2.
>
> **Q2: What is the difference between unsupervised reinforcement learning and Meta-RL, given that both seem to aim at rapidly solving new tasks?**
>
> **A:** Thanks a lot for the kind suggestion. As you have mentioned, both unsupervised RL and meta RL [1-3] focus on improving the generalization ability of RL agents, i.e., fast adapting to downstream tasks with limited steps. But they are different settings with the following differences.
>
> - Unsupervised RL pre-trains agents in **reward-free environments**, i.e., the environment without task rewards. Thus, we will design intrinsic rewards to pre-train agents to fully explore the environment. And the pre-trained agent needs to fast adapt to downstream tasks that it has not seen during the pre-training stage.
> - Meta RL pre-trains agents in **several sampled tasks from a task distribution $\mathcal{T}$**. And the pre-trained agent needs to fast adapt to test tasks, which are sampled from the same $\mathcal{T}$ but are not seen during the pre-training stage.
>
> Thanks again for your kind question. We have added the paragraph to describe the difference between unsupervised RL and meta RL in Appendix A in the revised version. We'd greatly appreciate it if you could provide us with more related works on meta RL.
>
> Reference:
>
> [1] Finn, Chelsea, Pieter Abbeel, and Sergey Levine. "Model-agnostic meta-learning for fast adaptation of deep networks." International conference on machine learning. PMLR, 2017.
>
> [2] Rakelly, Kate, et al. "Efficient off-policy meta-reinforcement learning via probabilistic context variables." International conference on machine learning. PMLR, 2019.
>
> [3] Zintgraf, Luisa, et al. "Varibad: Variational bayes-adaptive deep rl via meta-learning." Journal of Machine Learning Research 22.289 (2021): 1-39.

---

### Official Review · Reviewer_KGEL · 2025-11-04

**Soundness:** 3
**Presentation:** 2
**Contribution:** 3
**Rating:** 6
**Confidence:** 4

**Summary:**

-The paper introduces the Exploratory Diffusion Model (ExDM) to address the exploration bottleneck in Unsupervised Reinforcement Learning (URL).

-Unlike prior methods that use simple policies, ExDM leverages the superior expressive power of diffusion models to accurately model the complex and heterogeneous state distributions collected during exploration. A diffusion model ($\epsilon_{\theta^{\prime}}$) is trained on the replay buffer's state distribution. A novel score-based intrinsic reward ($R_{score}$) is calculated from this model's loss (its inability to fit a state), which guides the agent to under-visited regions. To ensure efficiency, a simple Gaussian behavior policy ($\pi_g$) is trained to maximize this $R_{score}$ and is used for fast data collection, avoiding slow diffusion sampling.
The pre-trained Gaussian policy ($\pi_g$) can be fine-tuned on downstream tasks using standard RL algorithms (like DDPG).

**Strengths:**

- The author addressed that this is the first work to successfully integrate diffusion models into the unsupervised exploration phase of RL. The concept of using the diffusion model's density estimation loss as the intrinsic reward is a significant contribution over prior reward mechanisms (like RND or ICM).
- It was impressed that the decoupled training scheme (fast Gaussian actor, slow diffusion reward-calculator) is a clever and practical solution to the primary obstacle of using generative models in online RL: slow sampling speed.
- I think that the paper provided a novel, non-trivial algorithm for fine-tuning the diffusion policy itself, complete with a formal proof of optimality (Theorem 4.2). This goes beyond just using the model as a static prior.
- Overall, the method's superior performance is not marginal. Its experiments dramatically outperform all baselines in complex exploration tasks (e.g., Fig. 2, where baselines get stuck and ExDM covers the entire maze) and shows consistent SOTA results across all aggregate metrics in URLB (Fig. 3).

**Weaknesses:**

- There is a limitation in terms of performance gap: The paper's own experiments (Fig. 3) show that fine-tuning the simple Gaussian policy ($\pi_g$) actually achieves better final performance than the proposed new, complex diffusion policy fine-tuning algorithm (Algorithm 2). The reason should be explained and analyzed intensively. Compared with Fig. 3(a) and (b), the expert normalized scores of the proposed algorithm in Fig. 3(c) were small.
- The authors stated that the performance degradation may be due to limited interaction timesteps during fine-tuning. The advanced works to overcome this problem should be discussed further.
- While their new fine-tuning method (Algorithm 2) is a novel contribution, it is not yet fully optimized and is outperformed by a simpler, standard approach such as DDPG.
Therefore, it is expected that the paper's primary strength lies in its pre-training exploration (which produces a superior Gaussian policy) rather than its diffusion policy fine-tuning performance. This mechanism could be considered in this discussion of this paper.

**Questions:**

Please, refer to the weakness and answer against my concerns.

---

> ### Author Response · Authors · 2025-11-20
> **Thanks for your supportive comments**
>
> Thanks a lot for your kind review and supportive comments. We'd like to address your concerns below:
>
> **W1-W2: about the performance of fine-tuning the diffusion policy and advanced works to overcome this problem**
>
> **A:** We thank the reviewer for the kind suggestion to discuss the diffusion policy fine-tuning performance of ExDM and advanced works addressing sample efficiency during fine-tuning. As you have mentioned, we have briefly discussed in Sec. 4.1 that the diffusion policy fine-tuning performance of ExDM is slightly worse than the Gaussian policy fine-tuning performance of ExDM. Our initial explanation in the manuscript pointed to the limited interaction steps during fine-tuning. Below, we will explain it in detail.
>
> Although diffusion policies have been proven as expressive models for capturing diverse offline datasets, online fine-tuning diffusion policies still face many challenges and are a key issue of concern for the community, for example, the instability caused by the multi-step sampling and the lack of closed-form probability calculation.
> Consequently, the diffusion policy inherently requires more interaction data to fine-tune effectively and may converge more slowly in a low-data regime (limited interaction steps).
> Although many recent works hope to address these challenges and improve the efficiency of online training diffusion policies, they mainly consider two settings: **online training diffusion policies from scratch with enough timesteps** or **online fine-tuning diffusion policies pre-trained from an offline dataset sampled from the same task**.
> In this work, we propose a more challenging setting for fine-tuning diffusion policies, i.e., we need to **online fine-tune diffusion policies to downstream tasks that are unseen during pre-training with limited timesteps**.
> And our experiments demonstrate that existing online diffusion methods perform poorly in this challenging setting, and ExDM outperforms these baselines, i.e., we hope to emphasize that
>
> - The performance of the diffusion policy fine-tuning of ExDM outperforms existing diffusion policy fine-tuning methods such as IDQL and DIPO.
> - The diffusion policy fine-tuning performance of ExDM is better than or comparable with the Gaussian policy fine-tuning performance of existing unsupervised RL methods like RND or CIC.
>
> Furthermore, we have added a paragraph in Appendix A of the revised PDF discussing relevant lines of work aimed at improving fine-tuning efficiency, like more energy-guided sampling techniques [1], more efficient diffusion sampling techniques (e.g., distillation, deterministic sampling) [2], and so on. We hope our work and this challenging setting can inspire more research on online fine-tuning diffusion policies in unsupervised RL. Thanks again for your kind suggestions. This discussion will better position our work within the broader literature and outline clear future research directions.
>
> **W3: About the contribution of the diffusion online fine-tuning algorithm in ExDM**
>
> **A:** We appreciate the reviewer for recognizing our contribution in unsupervised pre-training by first introducing diffusion models into unsupervised exploration. Below, we hope to clarify our contribution to the diffusion policy fine-tuning method of ExDM.
> As we have mentioned above, online fine-tuning diffusion policies still face several challenges, especially in the challenging setting of unsupervised RL.
> Consequently, one of our contributions in diffusion policy fine-tuning is to **formulate the online fine-tuning objective for unsupervised RL as Eq. 9** and propose an alternating optimization framework, of which **the optimality is guaranteed by Theorem 4.2**.
> Moreover, we extend our framework with a practical algorithm for diffusion policy fine-tuning and extensive experiments in Fig. 3 (c) show that ExDM outperforms existing diffusion policy fine-tuning baselines.
>
> Reference:
>
> [1] Liu, Xu-Hui, et al. "Energy-guided diffusion sampling for offline-to-online reinforcement learning." arXiv preprint arXiv:2407.12448 (2024).
>
> [2] Wang, Zhendong, et al. "One-step diffusion policy: Fast visuomotor policies via diffusion distillation." arXiv preprint arXiv:2410.21257 (2024).

---

### Author Response · Authors · 2025-11-20
**We have update our PDF following all your constructive comments**

We want to thank all the reviewers for their detailed comments and supportive review, including the novelty of first introducing diffusion models into unsupervised RL (Reviewer KGEL, UWWM); sufficient theoretical analyses and proof (Reviewer KGEL, ueyP, UWWM); superior performance across various settings (Reviewer KGEL, ueyP, dZCp, UWWM); and clear presentation that is easy to follow (Reviewer dZCp, UWWM).

Moreover, reviewers have posted several suggestions and concerns that are insightful and important for us to polish this work. We have provided a detailed response to each question in the corresponding rebuttal and made a number of changes in the revised PDF to address these suggestions and concerns. A summary of the modifications made is below (these modifications are in the revised PDF with highlighted  blue):

- Adding two entropy-based baselines, APT and APS, of URLB in Table 3 of Appendix D.5.
- Adding a detailed discussion of the difference between meta RL and unsupervised RL in Appendix A.
- Adding related works on the online fine-tuning diffusion policies in Appendix A.
- Adding the heatmap of maze2d of MEPOL and RE3 in  Fig.2.
- Changing the colors in Fig. 3 (b) to keep them the same as Fig. 3 (a), which will be more readable.
- Fixing several typos in the revised version of the paper, like adjusting Fig. 1 to make it more readable.

---

### Author Response · Authors · 2025-12-03
**Brief Summary of Rebuttal**

Dear AC (and SAC), ﻿

We sincerely express our gratitude to you and all the reviewers for your great efforts in reviewing our paper. To facilitate the evaluation of our work, we hereby provide a concise summary of the rebuttal.

First, we want to thank all reviewers for their detailed comments and supportive review (**all 4 reviewers gave a Rating of 6 with confidence 3 or 4 at the beginning**). Their positive reviews of this paper include:

- **The novelty of first introducing diffusion models into unsupervised RL** ("this is the first work to successfully integrate diffusion models into the unsupervised exploration phase of RL" by the reviewer KGEL, " While diffusion policies are not new, and diffusion models have been used to express various data distributions, their application to URL is novel to the best of my knowledge" by the reviewer UWWM)
- **Sufficient theoretical analyses and proof** ("I think that the paper provided a novel, non-trivial algorithm for fine-tuning the diffusion policy itself, complete with a formal proof of optimality" by the reviewer KGEL, "Sufficient theoretical proof" by the reviewer ueyP, "I consider the theoretical contributions of this work significant and valuable to the community" by the reviewer UWWM)
- **Superior performance across various settings** ("the method's superior performance is not marginal" by the reviewer KGEL, "The figure indicates consistent improvements over strong unsupervised exploration baselines in URL, in cross-embodiment transfer, and when initializing diffusion policies" by the reviewer ueyP, "The performance seems to be very strong compared to baselines" by the reviewer dZCp, "The empirical evaluation also appears to follow good practice" of the reviewer UWWM)
- **Clear presentation that is easy to follow** ("The presentation is clear and easy to follow" by the reviewer dZCp, "the writing and presentation of this work of a high quality" by the reviewer UWWM).

Moreover, reviewers have posted several insightful and important suggestions and concerns that will help us further refine this work. We have provided a detailed response to each question in the corresponding rebuttal, and we have made a number of changes in the revised PDF to address these suggestions and concerns. A summary of the modifications made is below (these modifications are in the revised PDF with highlighted blue):

- Adding two entropy-based baselines, APT and APS, of URLB in Table 3 of Appendix D.5.
- Adding a detailed discussion of the difference between meta RL and unsupervised RL in Appendix A.
- Adding related works on the online fine-tuning diffusion policies in Appendix A.
- Adding the heatmap of maze2d of MEPOL and RE3 in Fig.2.
- Changing the colors in Fig. 3 (b) to keep them the same as Fig. 3 (a), which will be more readable.
- Fixing several typos in the revised version of the paper, like adjusting Fig. 1 to make it more readable.

During the response stage, the reviewer **UWWM** has mentioned that all the concerns are fully addressed (especially about the integration and relevance of our diffusion policy approach) and has **raised the score to 8**.

Thank you very much again for your time in reviewing our work.

Sincerely, Authors

---

### Meta-Review · Area_Chair_agHp · 2026-01-04

**Summary:**

The reviewers noted a few shortcomings, including the main ones:
1) There is a limitation in terms of performance gap.
2) Algorithm 2 is not yet fully optimized and is outperformed by a simpler, standard approach such as DDPG.
3) APT and APS by Liu et al. should be included in the baseline.
4) What the diffusion policy of ExDM truly adds to the method.
5) Comparable effort to be spent on tuning hyperparameters across all approaches.

**Reviewer Concerns:**

The authors addressed all the comments made, namely:
1) The performance of fine-tuning: authors have added a paragraph in Appendix A of the revised PDF discussing relevant lines of work aimed at improving fine-tuning efficiency.
2) APT and APS: authors have supplemented APT and APS as baselines in fine-tuning the downstream tasks in URLB.
3) Contribution: during unsupervised exploration, ExDM introduces diffusion models to capture the diverse and heterogeneous trajectories and also ExDM pre-trains the diffusion policy to perform diversity.
4) Hyperparameters of baselines: authors take the same hyperparameters for both Maze2d environments for all baselines and ExDM.

The novelty and technical significance of the work were confirmed during the rebuttal.

**Reviewer Scores:**

1) Reviewer KGEL (score 6) would most likely have left his initial score.
2) Reviewer ueyP (score 6) would most likely have left his initial score.
3) Reviewer dZCp (score 6) would most likely have left his initial score.
4) Reviewer UWWM (score 6) confirmed explicitly that he would raise his initial score.

---

### Decision · Program_Chairs · 2026-01-26

Accept (Oral)